# Autocrine TGF-β-positive feedback in profibrotic AT2-lineage cells plays a crucial role in non-inflammatory lung fibrogenesis

Yasunori Enomoto [1,2], Hiroaki Katsura[1], Takashi Fujimura[1,3], Akira Ogata[1], Saori Baba[1], Akira Yamaoka [1], Miho Kihara[4], Takaya Abe [4], Osamu Nishimura [5], Mitsutaka Kadota[5], Daisuke Hazama[6], Yugo Tanaka[7], Yoshimasa Maniwa[7], Tatsuya Nagano[6] & Mitsuru Morimoto [1] ✉

The molecular etiology of idiopathic pulmonary fibrosis (IPF) has been extensively investigated to identify new therapeutic targets. Although anti-inflammatory treatments are not effective for patients with IPF, damaged alveolar epithelial cells play a critical role in lung fibrogenesis. Here, we establish an organoid-based lung fibrosis model using mouse and human lung tissues to assess the direct communication between damaged alveolar type II (AT2)-lineage cells and lung fibroblasts by excluding immune cells. Using this in vitro model and mouse genetics, we demonstrate that bleomycin causes DNA damage and activates p53 signaling in AT2-lineage cells, leading to AT2-to-AT1 transition-like state with a senescence-associated secretory phenotype (SASP). Among SASP-related factors, TGF-β plays an exclusive role in promoting lung fibroblast-to-myofibroblast differentiation. Moreover, the autocrine TGF-β-positive feedback loop in AT2-lineage cells is a critical cellular system in non-inflammatory lung fibrogenesis. These findings provide insights into the mechanism of IPF and potential therapeutic targets.

Idiopathic pulmonary fibrosis (IPF) is the most common type of interstitial lung disease (ILD) and is a chronic progressive disease with a poor prognosis of less than 5 years after diagnosis[1–3]. Molecular details of the etiology of IPF have been investigated to develop a new treatment approach, as there are only two approved anti-IPF drugs, nintedanib and pirfenidone; neither of which can completely halt the disease progression[1]. The hallmarks of IPF pathogenesis are epithelial damage and abnormal activation of fibroblasts for proliferation and differentiation into myofibroblasts, leading to the accumulation of excessive extracellular matrix proteins and scar formation in parenchymal tissue. These mesenchymal tissue malformations ultimately cause alveolar structural destruction, decreased lung compliance, and disrupted gas exchange function[4–6]. Although the pathogenic significance of immune cells, such as monocytes/macrophages, neutrophils, eosinophils, and several types of T cells in lung fibrosis has been reported[7,8], the role of inflammation in IPF is controversial because anti-inflammatory treatment is not effective or rather harmful for patients with IPF according to the PANTHER-IPF clinical trial[9].

[1]Laboratory for Lung Development and Regeneration, RIKEN Center for Biosystems Dynamics Research, 2-2-3 Minatojima-minamimachi, Chuo-ku, Kobe 650-0047, Japan. [2]Department of Regenerative and Infectious Pathology, Hamamatsu University School of Medicine, 1-20-1 Handayama, Higashi-ku, Hamamatsu 431-3192, Japan. [3]Department of Drug Modality Development, Osaka Research Center for Drug Discovery, Otsuka Pharmaceutical Co., Ltd., 5-1-35 Saitoaokita, Minoh 562-0029, Japan. [4]Laboratory for Animal Resources and Genetic Engineering (LARGE), RIKEN Center for Biosystems Dynamics Research, 2-2-3 Minatojima-minamimachi, Chuo-ku, Kobe 650-0047, Japan. [5]Laboratory for Phyloinformatics, RIKEN Center for Biosystems Dynamics Research, 2-2-3 Minatojima-minamimachi, Chuo-ku, Kobe 650-0047, Japan. [6]Division of Respiratory Medicine, Department of Internal Medicine, Kobe University Graduate School of Medicine, 7-5-2 Kusunoki-cho, Chuo-ku, Kobe 650-0017, Japan. [7]Division of Thoracic Surgery, Kobe University Graduate School of Medicine, 7-5-2 Kusunoki-cho, Chuo-ku, Kobe 650-0017, Japan. ✉e-mail: mitsuru.morimoto@riken.jp

Additionally, compared to those of patients with non-IPF lung fibrosis, such as fibrotic nonspecific interstitial pneumonia and collagen vascular disease-associated lung fibrosis, lungs of patients with IPF show fewer inflammatory pathologies despite more severe fibrosis[2,6,10,11]. Therefore, in clinical situations, inflammatory events may not be a good target for the treatment of IPF, and the non-inflammatory fibrosis process is worth evaluating.

The alveolar epithelium is composed of two types of epithelial cells: alveolar type I (AT1) cells, which are responsible for gas exchange, and alveolar type II (AT2) cells, which produce surfactants and function as tissue stem cells. If the alveolar epithelium is damaged, AT2 cells undergo self-renewal and differentiate into AT1 cells to regenerate lost alveolar epithelial cells[12]. Currently, there is growing evidence that non-inflammatory disorders of AT2 cells can cause lung fibrosis. Mutations in genes predominantly expressed in AT2 cells, such as surfactant protein C (SFTPC), surfactant protein A2, and ABCA3, have been associated with human lung fibrosis, including IPF[13]. In mouse models, dysfunction/stress of AT2 cells serves as an initiating event in lung fibrosis. Induction of telomerase deficiency[14], *Sin3a* knockout[15], endoplasmic reticulum stress[16,17], dysregulated autophagy[18], and mechanical stretching stress[19] in AT2-lineage cells result in the development of lung fibrosis; however, whether there is direct communication between dysfunctional AT2 cells and (myo) fibroblasts remains unclear.

The mechanism of non-inflammatory AT2-derived lung fibrogenesis has not been investigated in detail because of technical limitations. When using only in vivo fibrosis model, even with an *Sftpc-creER* driver that can genetically and specifically manipulate AT2 cells, the influence of immune cells is not completely excluded unless the lung tissue is isolated from the animal. Additionally, this in vivo fibrosis formation requires considerable time over several weeks, regardless of the induction methods, which makes it difficult to estimate whether AT2 cells directly or indirectly induce fibrosis. Therefore, it is desirable to develop a new experimental method to model lung fibrosis with minimal cell components such as profibrotic-stimulated AT2-lineage cells and naïve lung fibroblasts.

In this study, we aimed to clarify the causal relationship among epithelial injury, inflammation, and myofibroblast differentiation using an in vitro lung fibrosis organoid model for mice and humans, which recapitulates cell-cell interaction between damaged AT2-lineage cells and lung fibroblasts. We further modulated p53 and TGF-β signaling in vivo and in vitro, showing that the TGF-β-positive feedback loop in AT2-lineage cells is a critical cellular system in lung fibrogenesis. These data demonstrate that the initial induction of non-inflammatory lung fibrosis, including IPF, could be regulated by direct communication between damaged AT2-lineage cells and lung fibroblasts with no contribution from immune cells.

## Results

### Ex vivo and in vitro lung fibrosis models recapitulating inflammation-independent myofibroblast differentiation

To elucidate the temporal sequence of lung fibrosis, we generated a BLM-induced lung fibrosis model using an *Acta2-DsRed* mouse line, which exhibits fibroblast-to-myofibroblast differentiation by expressing DsRed fluorescent protein, and examined the fibrosis phenotype on days 0 to 21 (Fig. 1a). We found that epithelial injury, defined as positivity for the DNA double-strand break marker γH2AX, was the initial event after BLM injection (day 2) (Fig. 1b), followed by the appearance of infiltrated immune cells (days 4–7) (Fig. 1c and Supplementary Fig. 1a). Thereafter, DeRed⁺ myofibroblasts (day 7–) and collagen accumulation (day 14–) became evident (Fig. 1a, d).

To model epithelial injury and inflammation-independent fibroblast-to-myofibroblast differentiation, we used two culture systems: ex vivo lung-lobe culture (Fig. 1e, see Methods) and in vitro feeder-free alveolar organoid culture assays (Fig. 1h). On ex vivo

culture day 2, γH2AX-expressing damaged cells appeared in BLM-treated lobes and frequently merged with the AT2 cell marker, SFTPC (Fig. 1e and Supplementary Fig. 1b). On day 12, SFTPC and γH2AX expression was lost, and DsRed⁺ myofibroblasts were increased in the subpleural area of BLM-treated lobes (Fig. 1e). Quantitative reverse transcription PCR (qRT-PCR) revealed a sixfold increase in *Acta2* expression in BLM-treated lobes (Fig. 1f), indicating that BLM-mediated AT2 cell dysfunction may induce myofibroblasts without inflammation. Hematopoietic cells in the ex vivo culture were measured using FACS with an anti-CD45 antibody, confirming that the amount of hematopoietic cells was negligible on day 12 (Fig. 1g). To exclude the possibility that BLM directly induces fibroblast-to-myofibroblast differentiation, we performed a primary culture of FACS-sorted PDGFRα⁺EPCAM⁻CD45⁻CD31⁻LYVE1⁻CD146⁻ lung fibroblasts and confirmed that BLM administration to fibroblasts did not increase *Acta2* expression (Supplementary Fig. 1c). These data suggest that BLM-induced DNA damage in AT2 cells, and not in fibroblasts, could be the initial trigger for inflammation-independent fibroblast-to-myofibroblast differentiation.

Subsequently, we developed an alveolar organoid culture system for a lung fibrosis model (Fig. 1h and Supplementary Fig. 1d, see Methods), which is a constructive approach that allowed us to determine the minimum cell components to induce fibroblast-to-myofibroblast differentiation. Similar to the in vivo and ex vivo BLM-treated lung lobes (Fig. 1b, e), the majority of cells in BLM-treated alveolar organoids showed γH2AX expression (Fig. 1i, j), indicating epithelial DNA damage. To assess the capacity of AT2 cells to induce fibroblast-to-myofibroblast differentiation, we co-cultured BLM-treated alveolar organoids with naïve lung fibroblasts isolated from *Acta2-DsRed* mice at 24 h after BLM treatment (Fig. 2a and Supplementary Fig. 2, see Methods). If fibroblasts differentiate into myofibroblasts, DsRed fluorescence is observed in the surrounding organoid-containing gel. As expected, fluorescence microscopy revealed an increase in DsRed fluorescence in fibroblasts co-cultured with BLM-treated alveolar organoids (Supplementary Movies 1 and 2). These DsRed⁺ myofibroblasts co-cultured with BLM-treated organoids were more and larger than those co-cultured with control organoids, although the cell number itself did not differ (Fig. 2b), indicating that fibroblast-to-myofibroblast differentiation was directly induced by BLM-treated AT2 cells. Thereafter, we evaluated the sensitivity of the *Acta2-DsRed* reporter under our culture conditions with TGF-βs (a mixture of recombinant TGF-β1, -β2, and -β3) as a potent profibrotic factor at different concentrations (Fig. 2c), and determined that the reporter assay was able to detect recombinant TGF-βs as low as 1 pg mL⁻¹ and showed a dose-dependent increase within 100 pg mL⁻¹ (Fig. 2d). Collectively, we successfully established a protocol for alveolar organoid culture in a lung fibrosis model with sufficient sensitivity to detect low-dose profibrotic factors, which can recapitulate inflammation-independent epithelial-mesenchymal profibrotic interactions.

### Profibrotic significance of p53 signaling in AT2-lineage cells

To explore the cell signaling pathways that enforce the profibrotic state of AT2 cells downstream of DNA damage, we performed bulk RNA-seq of BLM-treated and non-treated alveolar organoids from four independent mice. Pathway analysis revealed that the p53 signaling pathway, which is a well-known stress response pathway regulating cellular senescence, cell cycle arrest, and apoptosis was markedly enriched in BLM-treated organoids (Fig. 3a, b). Treatment with BLM impeded organoid growth, with more apoptotic cells and fewer proliferative cells (Supplementary Fig. 3a–d). In contrast, canonical AT2 or AT1 markers, such as *Sftpc*, *Abca2*, *Pdpn*, and *Hopx*, decreased, suggesting that the cell state of AT2 cells is altered by DNA damage (Fig. 3b). The expression of the *Trp53* gene was not changed; however, the p53 protein level and its acetylated form increased in BLM-treated

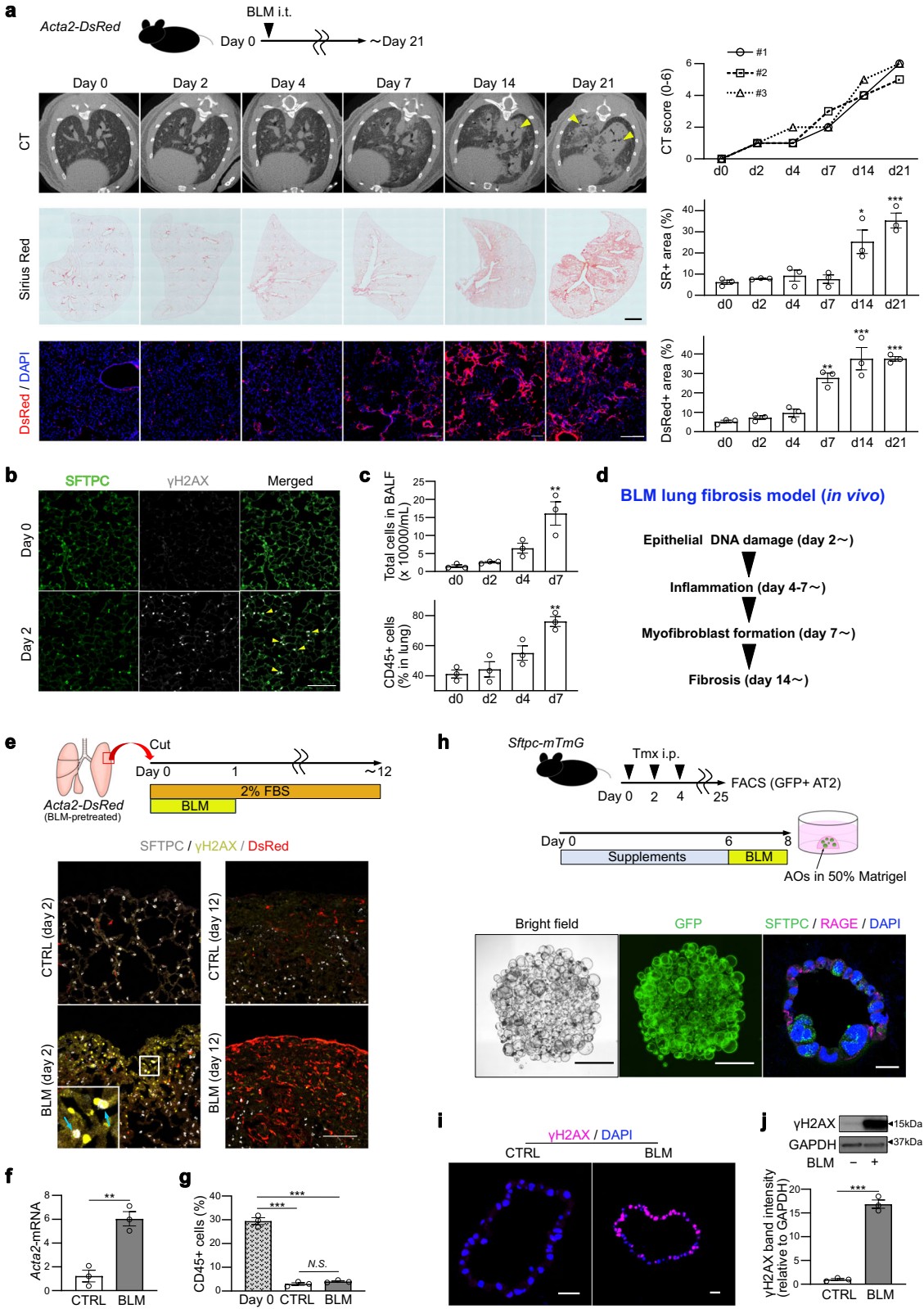

organoids (Fig. 3c), suggesting the activation of p53 signaling. These damaged organoids were well stained with p21^Waf1/Cip1 and senescence-associated β-galactosidase, although other major senescent markers, such as p16^INK4A and p19^ARF, were not increased (Fig. 3c–e and Supplementary Fig. 3e). Taken together, the stress of DNA damage in AT2-lineage cells could stimulate the p53 signaling pathway via post-translational modifications and enforce the senescence state of AT2 cells.

Recently, the role of p53 signaling in AT2-lineage cells for fibrosis development has been proposed[15,20]; however, the mechanism underlying fibroblast-to-myofibroblast differentiation induced by p53-active AT2 cells remains elusive. We first used alveolar organoids cultured with Nutlin-3a, a chemical activator of p53 signaling, instead of BLM for 24 h, and subsequently co-cultured with *Acta2-DsRed* lung fibroblasts for the lung fibrosis model (Fig. 3f). Notably, an increase in the DsRed-positive area, but not in the cell number, surrounding

**Fig. 1 | BLM-induced lung fibrosis model with mice, ex vivo culture, and in vitro organoid culture. a** Serial assessment of BLM-induced lung fibrosis model in vivo using micro-CT (*upper*), Sirius Red (SR) staining (*middle*; scale bar: 1 mm), and DsRed reporter (*lower*; scale bar: 200 μm). Triangles on micro-CT images indicate fibrosis areas. Right graphs are the quantifications. CT score in each timing is derived from three independent mice. Other data represent the mean ± SEM of results obtained from three independent mice in each timing. *$P < 0.05$, **$P < 0.01$, ***$P < 0.001$ compared to day 0. **b** Lung sections on days 0 and 2 stained for SFTPC, γH2AX (scale bar: 100 μm). Triangles indicate γH2AX, SFTPC-double positive cells. **c** The number of total cells in BALF (*upper*) and percentages of CD45+ cells in whole lung lysates (*lower*). Data represent the mean ± SEM of results obtained from three independent mice. **$P < 0.01$ compared to day 0. **d** Summary scheme of BLM-induced lung fibrosis model in vivo. **e** Diagram of ex vivo lung-lobe culture and sections of the cultured lung tissue stained for SFTPC, γH2AX, and DsRed

fluorescence (scale bar: 100 μm). Arrows indicate γH2AX-positive AT2 cells. **f** qRT-PCR of ex vivo cultured lungs. Data represent the mean ± SEM of the results obtained from three independent mice. **$P < 0.01$. **g** Evaluation of CD45+ cells by FACS for ex vivo cultured lung on day 12. Data represent the mean ± SEM of the results obtained from three independent mice. ***$P < 0.001$ compared to day 0. **h** Diagram of alveolar organoid (AO) culture and in vitro BLM treatment, images of untreated AOs in bright-field and GFP fluorescence on day 8 (scale bar: 1 mm), and sections stained for SFTPC and RAGE (scale bar: 20 μm). **i** Sections of BLM-treated AOs stained for γH2AX (scale bar: 20 μm). **j** Western blotting of CTRL- and BLM-AOs for γH2AX and GAPDH and the quantification. The lanes were run on the same gel. Data represent the mean ± SEM of results obtained from three independent mice. All statistical analyses are evaluated by unpaired two-tailed Student's *t*-test or one-way analysis of variance with Bonferroni correction, as appropriate.

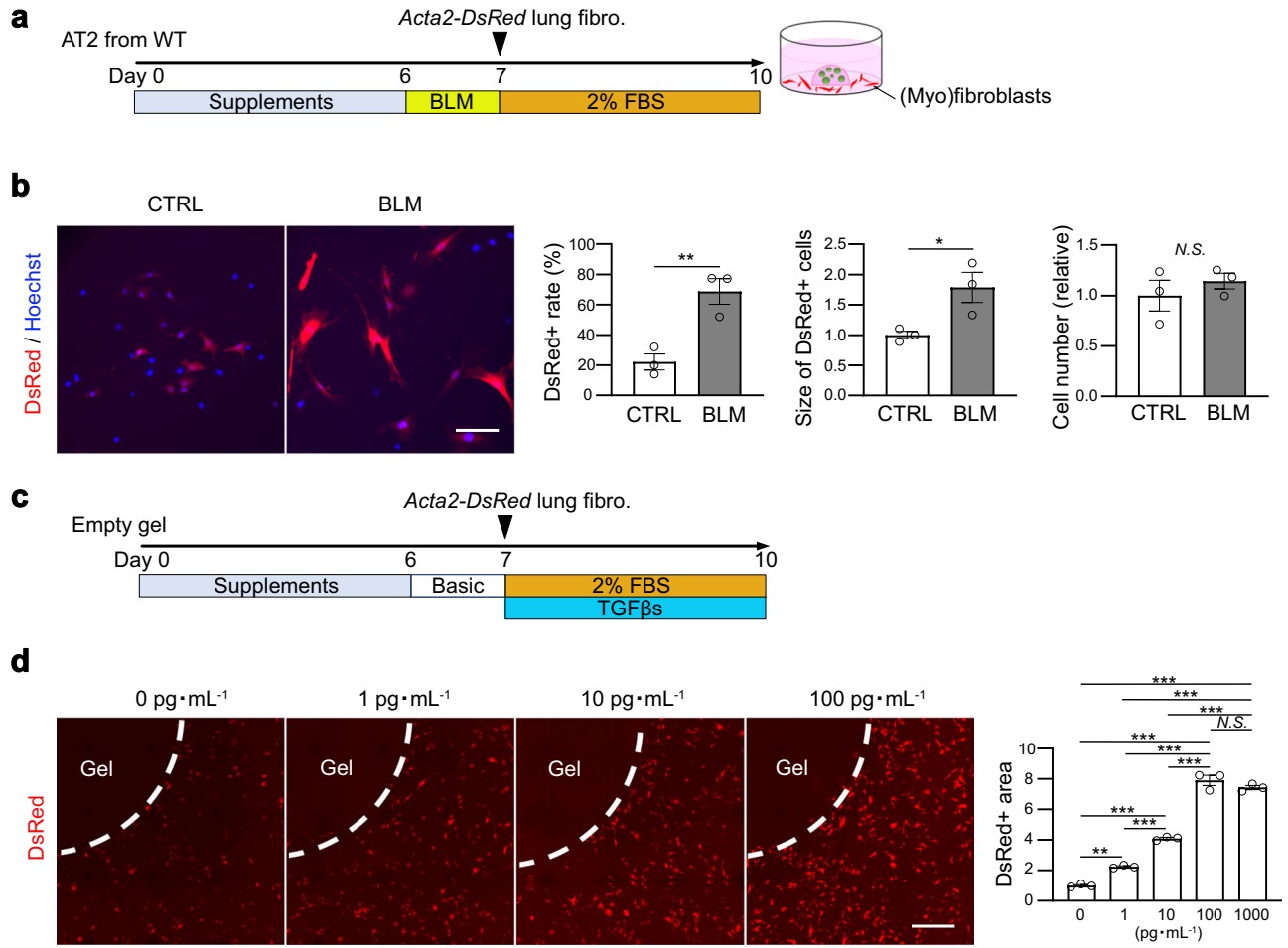

**Fig. 2 | Fluorescence reporter system for differentiating myofibroblasts activated by profibrotic alveolar organoids. a** Diagram of co-culture using AOs from wild-type (WT) mice with primary lung fibroblasts from *Acta2-DsRed* mice. **b** Representative images of co-cultured (myo)fibroblasts (scale bar: 100 μm) and quantification of the number and the relative size of DsRed+ myofibroblasts and the relative number of total fibroblasts around the AO-containing gel. Data represent the mean ± SEM of results obtained from three independent mice. *$P < 0.05$,

**$P < 0.01$ (unpaired, two-tailed Student's *t* test). **c** Diagram of sensitivity analysis using *Acta2-DsRed* lung fibroblasts using mixed TGF-β1/β2/β3. **d** Representative images of (myo)fibroblasts (scale bar: 1 mm) and quantification of the relative DsRed+ area around the gel. Data represent the mean ± SEM of results obtained from three independent mice. **$P < 0.01$, ***$P < 0.001$ (one-way analysis of variance with Bonferroni correction).

Nutlin-3a-treated organoids was detected, similar to that of BLM-treated ones, suggesting that activation of the p53 signaling pathway is sufficient for profibrotic AT2 cells. To confirm the profibrotic role of p53 in vivo, we generated BLM-induced lung fibrosis model animals using an AT2-specific p53 conditional knockout mouse line, *Sftpc$^{CreERT2}$; Trp53$^{flox/flox}$; Rosa26$^{mTmG}$* (hereafter p53-cKO). As expected, p53-cKO mice were more resistant to BLM-induced lung fibrosis than the control, showing less body weight loss (Supplementary Fig. 4a), less

collagen deposition, and milder mCT-based fibrotic changes on day 21 (Fig. 4a, b). Immunostaining for α-smooth muscle actin (αSMA, encoded by *Acta2*), GFP (AT2-lineage cells), and p21$^{Waf1/Cip1}$ (a senescence marker) showed fewer αSMA+ myofibroblasts and p21+GFP+ cells in the mutant lungs on day 7, suggesting that p53-cKO mice are resistant to the induction of fibroblast-to-myofibroblast differentiation and cellular senescence in AT2-lineage cells (Fig. 4a, b). We further performed ex vivo BLM-treated lobe culture and found that *Acta2* gene expression

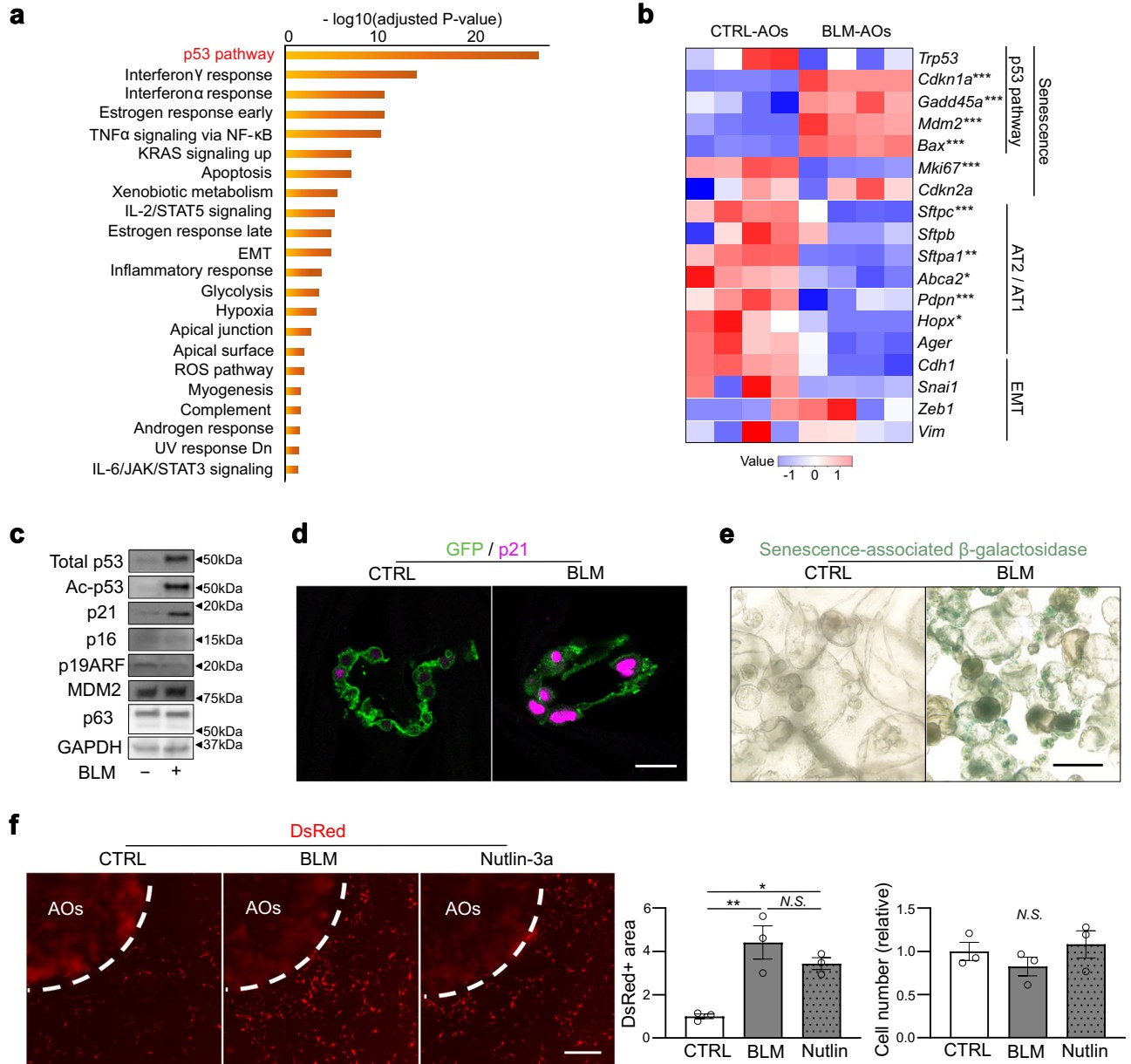

**Fig. 3 | BLM treatment activates the p53 pathway and profibrotic cellular senescence phenotype in alveolar organoids. a** Enriched pathways of bulk RNA-seq data of BLM-treated AOs using "MSigDB Hallmark 2020". A total of 2677 genes that were significantly upregulated in BLM-AOs (compared to CTRL-AOs) were evaluated. Data were obtained from four independent Tamoxifen-treated *Sftpc^{CreERT2}: Rosa26^{mTmG}* mice. The names of upregulated genes and normalized CPM values are included in Supplementary Data 2. *P* values are evaluated using the Fisher's exact test that assumes a binomial distribution and independence for probability of any gene belonging to any set. **b** Heatmap visualization of selected genes (senescence-, cell lineage-, and epithelial-mesenchymal transition [EMT]-related genes) via bulk RNA-seq of CTRL- and BLM-AOs. *FDR < 0.05, **FDR < 0.01, ***FDR < 0.001 (Benjamini-Hochberg method). **c** Representative images of protein expression of CTRL- and BLM-AOs via western blotting for total p53, acetylated p53 (Lys379), p21^{Waf1/Cip1}, p16, p19^{ARF}, MDM2, p63, and GAPDH. The samples were derived from the same experiment and gels/blots were processed in parallel. **d** Representative images of AO sections stained with anti-GFP and anti-p21 antibodies (scale bar: 20 μm). **e** Representative images of AOs stained with senescence-associated β-galactosidase (scale bar: 200 μm). **f** Representative images of co-cultured (myo)fibroblasts using AOs (from wild-type mice) treated with BLM (100 μM, 24 h) or Nutlin-3a (2 μM, 24 h) (*scale bar: 1 mm*), quantification of the relative DsRed⁺ area around the AO-containing gel, and the relative number of total fibroblasts. Data represent the mean ± SEM of results obtained from three independent mice. *P < 0.05, **P < 0.01 (one-way analysis of variance with Bonferroni correction).

was suppressed in p53-cKO lobes compared to that in the control, indicating that resistance to fibrosis in the mutant is not related to immune cells (Supplementary Fig. 4b). To directly demonstrate that p53-null AT2 cells are not profibrotic even after BLM treatment, alveolar organoid culture was performed using p53-null AT2 cells isolated from lungs of p53-cKO mice. *Acta2*-reporter activity was remarkably suppressed by p53 ablation in AT2 cells despite considerable DNA damage due to BLM (Fig. 4c, d). These data demonstrate that

p53 signaling in AT2-lineage cells plays an indispensable role in fibroblast-to-myofibroblast differentiation by regulating the interaction of AT2-lineage cells with lung fibroblasts.

It has been suggested that Sirtuin, a histone deacetylase, may be involved in susceptibility to lung fibrosis[21,22]. As SIRT1 is a negative regulator of p53 and as we observed a marginal decrease in SIRT1 protein levels in BLM-treated alveolar organoids (Supplementary Fig. 4c), we generated AT2-specific *Sirt1* conditional knockout and overexpression

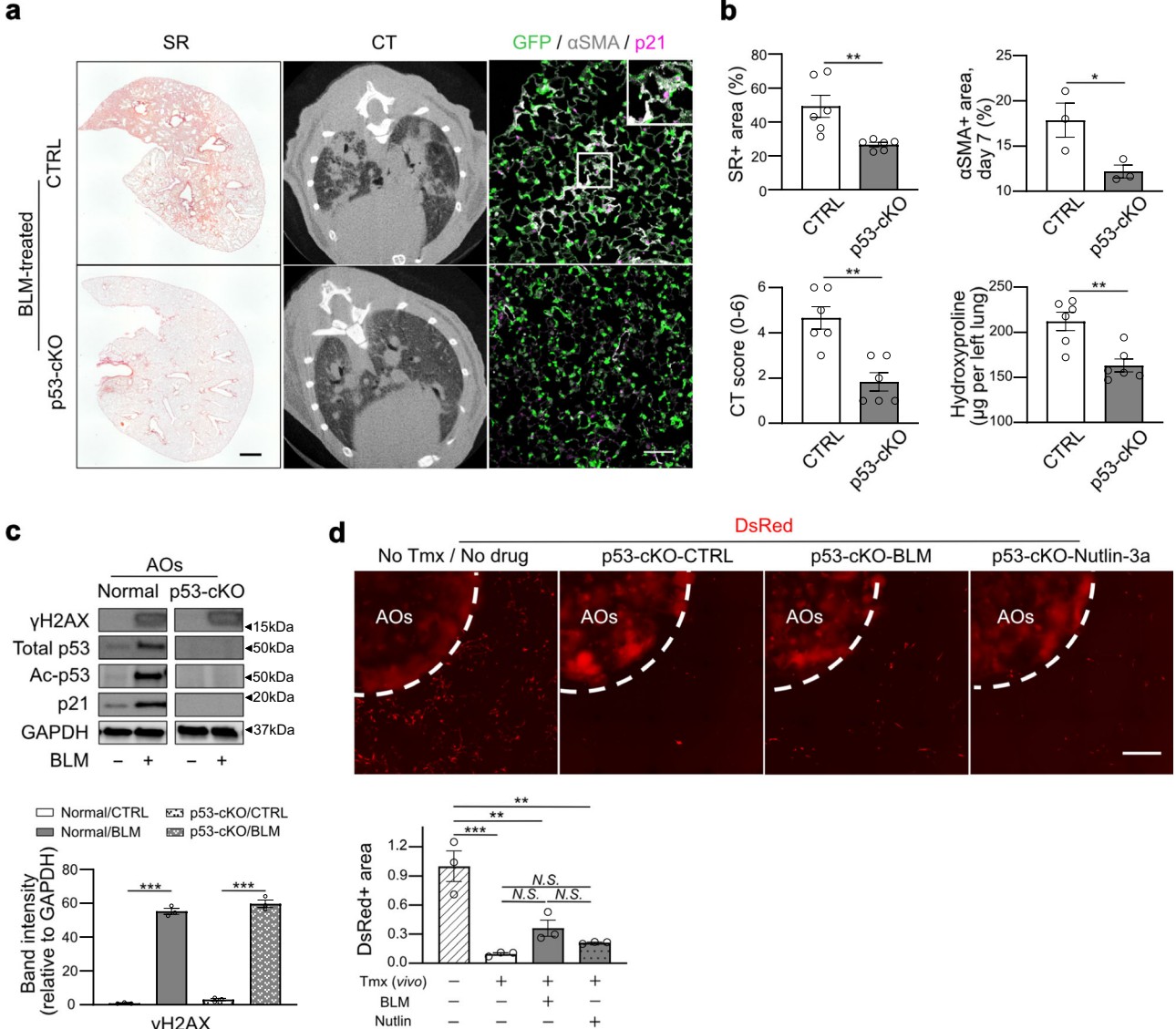

**Fig. 4 | Profibrotic significance of p53 signaling in AT2(-lineage) cells.**
**a** Representative images of the lung sections for Sirius Red (SR) staining (scale bar: 1 mm), lung mCT, and lung sections stained with anti-GFP, anti-p21, and anti-αSMA antibodies (scale bar: 100 μm). *Sftpc^CreERT2^; Rosa26^mTmG^* mice (CTRL) and *Sftpc^CreERT2^; Trp53^flox/flox^; Rosa26^mTmG^* mice (p53-cKO) were treated with BLM. **b** Quantification of SR staining (day 21, *n* = 6), mCT (day 21, *n* = 6), αSMA⁺ area (day 7, *n* = 3), and hydroxyproline in the left lungs (day 21, *n* = 6). Data represent the mean ± SEM of results obtained from three or six independent mice. *$P < 0.05$, **$P < 0.01$ (unpaired, two-tailed Student's *t* test). **c** Representative images of the protein expression of AOs from *Sftpc^CreERT2^; Trp53^flox/flox^; Rosa26^mTmG^* mice (p53-cKO) via western blotting for total p53, acetylated p53, γH2AX, and GAPDH. The lanes were run on the same

gel but were noncontiguous. The samples were derived from the same experiment and gels/blots were processed in parallel. The lower graph is the quantification of γH2AX expression. For p53 and p21, no bands were detected in p53-cKO-derived AO. Data represent the mean ± SEM of results obtained from three independent mice. ***$P < 0.001$ (unpaired, two-tailed Student's *t* test). **d** Representative images of co-cultured (myo)fibroblasts using p53-cKO AO treated with BLM (100 μM, 24 h) or Nutlin-3a (2 μM, 24 h) (scale bar:1 mm) and quantification of the relative DsRed⁺ area around the AO-containing gel. Data represent the mean ± SEM of results obtained from three independent mice. **$P < 0.01$, ***$P < 0.001$ (one-way analysis of variance with Bonferroni correction).

---

mouse lines (*Sftpc^CreERT2^; Sirt1^flox/flox^* and *Sftpc^CreERT2^; Rosa26^CAG-LSL-Sirt1-P2A-eGFP^*) (Supplementary Fig. 4d, e). To evaluate the role of SIRT1 in AT2-lineage cells during lung fibrosis, we intratracheally administered BLM to these mice. However, neither knockout nor overexpression of *Sirt1* influenced the severity of the lung fibrosis phenotype (Supplementary Fig. 4f), suggesting that SIRT1 in AT2-lineage cells is dispensable in the pathogenesis of BLM-induced lung fibrosis.

**TGF-β-positive feedback loop in AT2- lineage cell is crucial for lung fibrogenesis**
To identify the secretion factors expressed downstream of p53 signaling in AT2-lineage cells that directly induce fibroblast-to-

myofibroblast differentiation, we evaluated the expression of senescence-associated secretory phenotype (SASP) factors[23,24] in alveolar organoids two days after BLM treatment (culture day 9). We found that compared to BLM-treated normal alveolar organoids, BLM-treated p53-null organoids showed reductions in several SASP genes: *Tgfb1, Tgfb2, Tgfb3, Ccn2* (a gene of CTGF), *Pdgfa, Pdgfb, Vegfa, Vegfc, Csf2, Cxcl12,* and *Ccl3* (Fig. 5a). To validate the profibrotic capacity of these factors, we cultured *Acta2-DsRed* lung fibroblasts with recombinant proteins (10 ng/mL each). We found that only TGF-β isoforms (β1, β2, and β3) upregulated *Acta2* and *Col1a1*, although the expression of *Cthrc1*, another myofibroblastic marker[25], was enhanced by CTGF, PDGF, and TGF-β isoforms (Supplementary Fig. 5a). Using a BLM-

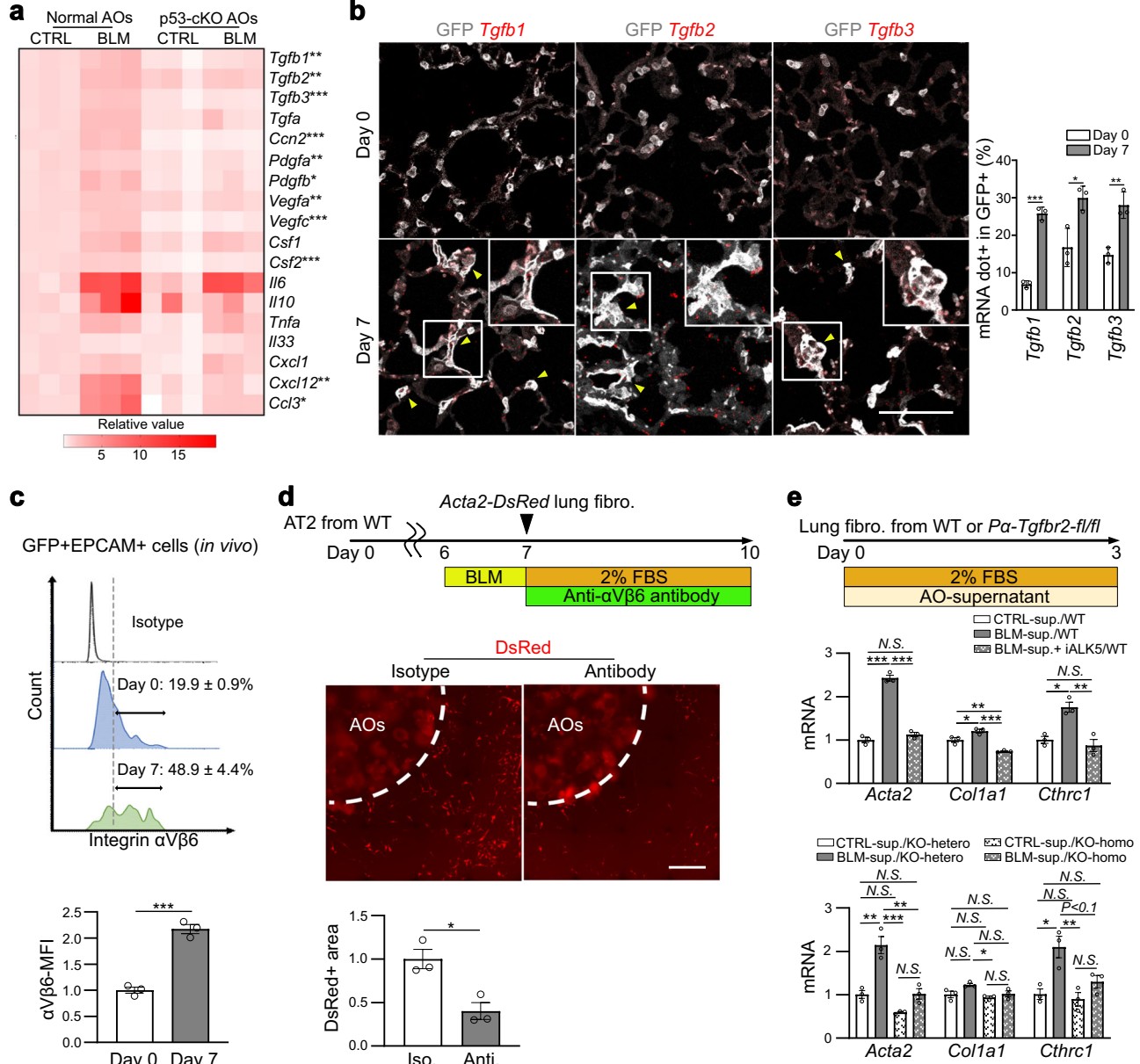

**Fig. 5 | TGF-β-related genes in BLM-treated AT2-lineage cells exclusively contribute to myofibroblast differentiation within SASP factors. a** Heatmap visualization of mRNA expression of SASP factors evaluated by qRT-PCR using normal AOs from *Sftpc*^CreERT2^*; Rosa26*^mTmG^ mice and p53-cKO AOs from *Sftpc*^CreERT2^*; Trp53*^flox/flox^*; Rosa26*^mTmG^ mice. AOs were treated with 100 μM BLM for 24 h, and harvested on culture day 9. Data were obtained from three independent mice. *P < 0.05, **P < 0.01, ***P < 0.001 compared between BLM-treated normal AOs and BLM-treated p53-cKO AOs (unpaired, two-tailed Student's t test). **b** Representative images of lung sections for in situ hybridization and quantification using *Sftpc*^CreERT2^*; Rosa26*^mTmG^ mice on days 0 and 7 after BLM injection in vivo. Yellow triangles indicate mRNA-dot-positive GFP+ AT2-lineage cells. Data represent the mean ± SEM of results obtained from three independent mice. *P < 0.05, **P < 0.01, ***P < 0.001 (unpaired, two-tailed Student's t test). (scale bar: 100 μm). **c** FACS panels with quantification of mean fluorescence intensity (MFI) for integrin αVβ6 using GFP+ AT2-lineage cells (days 0 and 7, in vivo). Data represent the mean ± SEM of results

obtained from three independent mice. ***P < 0.001 compared to day 0 (unpaired, two-tailed Student's t test). **d** Experimental set-up, representative images of co-cultured AOs and myofibroblasts (scale bar: 1 mm), and quantification of the relative DsRed+ area around the AO-containing gel using normal AOs treated with BLM (100 μM, 24 h) and subsequently with a neutralizing antibody against integrin αVβ6 (100 μg mL⁻¹, 72 h). Data represent the mean ± SEM of results obtained from three independent mice. *P < 0.05 (unpaired, two-tailed Student's t test). **e** Diagram of the experiments and comparison of mRNA expression levels by qRT-PCR for primary lung fibroblasts. Lung fibroblasts were isolated from wild-type mice (*upper panel*) or tamoxifen-treated transgenic mice (*lower panel*): *Pdgfra*^CreERT2^*; Tgfbr2*^flox/flox^ or *Pdgfra*^CreERT2^*; Tgfbr2*^flox/+^. These fibroblasts were treated with culture supernatants from normal AOs that were pre-treated with BLM (100 μM, 24 h) and then cultured for the next 72 h. Data represent the mean ± SEM of results obtained from three independent mice. *P < 0.05, **P < 0.01, ***P < 0.001 (one-way analysis of variance with Bonferroni correction).

induced lung fibrosis model, we confirmed increased gene expression of all TGF-β isoforms in AT2-lineage cells in vivo using PLISH, a single molecular RNA in situ hybridization[26] (Fig. 5b). Given that integrin αVβ6 activates latent TGF-β1 and -β3[27,28], we evaluated the ratio of integrin αVβ6+ cells in AT2-lineage cells using FACS. This quantitative

analysis determined that 20% of AT2-lineage cells expressed αVβ6 even on day 0, and increased to 50% on day 7, which was validated using mean fluorescence intensity analysis (Fig. 5c and Supplementary Fig. 5b). qRT-PCR also confirmed BLM or Nutlin-3a-induced increases in the expression of TGF-β isoforms and integrins in wild-type alveolar

organoids and non-significant changes in p53-null alveolar organoids (Supplementary Fig. 5c, d). Based on these observations, we expected that p53 signaling activates latent TGF-β in AT2-lineage cells by increasing their integrin αVβ6 expression. To test this hypothesis, we aimed to inhibit fibroblast-to-myofibroblast differentiation in our culture assay by adding a neutralizing antibody against integrin αVβ6. The neutralizing antibody successfully suppressed *Acta2*-reporter activity (Fig. 5d). These data are consistent with a previous study on Sin3a-cKO mice, in which p53-enriched AT2-lineage cells show upregulation of *Tgfb1*, *Tgfb3*, *Itgav*, and *Itgb6* (Supplementary Fig. 5e)[15].

To elucidate whether TGF-β signaling plays an exclusive role in fibroblast-to-myofibroblast differentiation in our alveolar organoid culture, we inhibited TGF-β signaling in lung fibroblasts using pharmacology and genetics. Lung fibroblasts were incubated with culture supernatants collected from the BLM-treated alveolar organoids. We found that the increase in the expressions of myofibroblast-related genes by the BLM-treated organoid supernatant was suppressed by the addition of SB431542, an ALK5 inhibitor (Fig. 5e). We further prepared Tgfbr2-null lung fibroblasts from *Pdgfra^CreERT2*; *Tgfbr2^flox/flox* mice and cultured them with BLM-treated organoid supernatants. Consistent with the pharmacological assay, Tgfbr2-null lung fibroblasts did not respond to the supernatant (Fig. 5e). Furthermore, to evaluate additional BLM-treated AT2-derived profibrotic factors, we cultured fibroblasts and treated with organoid supernatant with or without low-dose TGF-βs (10 pg mL⁻¹). As shown in Supplementary Fig. 5f, only additive, but not synergistic, changes in the expressions of myofibroblast-related genes were observed.

A time lag between p53 activation (day 7) and upregulation of profibrotic TGF-β-related genes (day 9) prompted us to evaluate whether TGF-β isoforms and integrins are direct targets of p53. Therefore, we performed ChIP-seq analysis using an anti-p53 antibody for BLM-treated alveolar organoids to determine the direct targets of p53. We observed peaks upstream p53 target genes, such as *Cdkn1a* (Fig. 6a). However, there were no apparent differences between control- and BLM-treated-alveolar organoids in the binding of p53 to TGF-β isoforms and integrins (Fig. 6a), suggesting that p53 does not directly induce these profibrotic factors. However, the direct p53 target genes identified by our ChIP-seq analysis included several TGF-β-related genes (Fig. 6b). We identified 329 genes that were the direct targets of p53 and upregulated in our RNA-seq data, including genes that have been reported to have a potential to activate TGF-β signaling, such as *Areg*[29], *Cdkn1a*[30], *Ltbp2*[31], *Pdgfc*[32], and *Tgfa*[33] (Fig. 6c and Supplementary Data 4). These data suggest that p53 indirectly upregulates profibrotic factors by inducing these candidate factors.

## Cell-autonomous TGF-β-positive feedback loop in AT2-lineage cells is crucial for lung fibrogenesis

Next, we investigated whether AT2-lineage cells were targets of TGF-β. We prepared a BLM-induced lung fibrosis model using *Sftpc^CreERT2*; *Rosa26^mTmG* mice to trace AT2-lineage cells during the development of fibrosis. Immunostaining showed phosphorylated SMAD2/3 in GFP⁺, Krt8⁺ AT2-lineage cells and αSMA⁺ myofibroblasts (triangles in Fig. 7a and Supplementary Fig. 6a), suggesting that the canonical TGF-β-SMAD pathway is activated in AT2-lineage cells in BLM-induced lung fibrosis. In addition, when using BLM-treated alveolar organoids, we confirmed the accumulation of phosphorylated SMAD2 on western blotting (Fig. 7b). As our feeder-free alveolar organoid assay contained only AT2-derived cells, TGF-β signaling was activated in an autocrine manner. To explore the pathogenic significance of autocrine TGF-β signaling in fibrosis, we generated an AT2-specific *Tgfbr2* conditional knockout mouse line, *Sftpc^CreERT2*; *Tgfbr2^flox/flox* (hereafter Tgfbr2-cKO), to disturb TGF-β signaling in AT2-lineage cells and prepared a BLM-induced lung fibrosis model. Similar to the phenotype of p53-cKO mice (Fig. 4), Tgfbr2-cKO mice showed resistance to BLM-induced lung fibrosis with less body weight loss (Supplementary Fig. 6b), less

deposition of collagen, smaller fibrotic lesion area on day 21, and fewer myofibroblasts on day 7 (Fig. 7c, d), which was consistent with the results of the ex vivo lung-lobe culture (Supplementary Fig. 6c) and a previous report using *Nkx2.1-cre* driver[34]. To observe the interaction between AT2 cells and lung fibroblasts, a co-culture assay was performed using Tgfbr2-null AT2 cells isolated from Tgfbr2-cKO lungs. We found that *Acta2*-reporter activity was comparable between BLM-treated and non-treated organoids of Tgfbr2-null AT2 cells (Fig. 7e), demonstrating that autocrine TGF-β signaling in AT2 cells plays a crucial role in fibroblast-to-myofibroblast differentiation in BLM-induced lung fibrosis. We then investigated whether TGF-β treatment enforces the profibrotic state of AT2 cells. In our alveolar organoid culture for the lung fibrosis model, exogenous TGF-β treatment increased myofibroblast formation (Fig. 7f and Supplementary Fig. 7a) and the expression of TGF-β-related genes including integrins was enhanced in AT2 cells (Supplementary Fig. 7b), which is also observed in other cell types as a TGF-β-integrin crosstalk[35]. However, these genes did not respond to BLM treatment in Tgfbr2-null alveolar organoids (Supplementary Fig. 7c). Furthermore, both an ALK5 inhibitor (SB431542) and a neutralizing antibody for integrin αVβ6 decreased the expression of the profibrotic gene set in BLM-treated wild-type alveolar organoids (Supplementary Fig. 7d). Collectively, TGF-β exposure in AT2 cells enhanced their profibrotic potential by inducing TGF-β-related gene expression, which further activated TGF-β signaling in AT2 cells in an autocrine manner. This autocrine positive-feedback loop of TGF-β signaling could amplify the profibrotic potential of AT2-lineage cells and may be a key cellular system in the development of lung fibrosis. In contrast, we found that treatment of fibroblasts with TGF-βs also resulted in upregulated expression of TGF-β isoforms, particularly *Tgfb1* (Supplementary Fig. 7e), indicating that the TGF-β positive-feedback loop is a common cell response not only in AT2 cells but also in fibroblasts, which may contribute to maintaining the profibrotic state in both cells and accelerate the TGF-β signal interaction even without immune cells.

## Autocrine TGF-β signaling is sufficient to convert AT2 cells into profibrotic AT2-to-AT1 transition-like cell state

Recently, several groups have reported a unique alveolar epithelial cell state called PATS/DATP/KRT8⁺ transitional cells (hereafter PATS cells), which are an intermediate cell population from the differentiation of AT2 to AT1 cells in the regenerating lung[20,36,37]. PATS cells demonstrate a feature of cellular senescence, including activation of p53 signaling. PATS cells are expected to be involved in lung fibrogenesis because their marker genes largely overlap with the major upregulated genes in the alveolar epithelium of mouse BLM-induced fibrotic lungs and human IPF lungs[20]. However, to date, no direct evidence showing the profibrotic function of PATS/PATS-like cells has been provided. To confirm this, we first examined whether AT2-lineage cells with an activated autocrine TGF-β signaling loop are equivalent to PATS cells. We quantified the expression of PATS- and TGF-β-related genes in integrin αVβ6-positive AT2-lineage cells as TGF-β-signaling-active cells because high expression of integrin αVβ6 can be a hallmark of the signaling loop. These cells were collected from BLM-treated lungs on day 7 and compared with integrin αVβ6-negative cells (Fig. 8a). As expected, most of these genes, including *Cldn4*, *Krt8*, *Tgfb1*, and *Tgfb2*, were upregulated in integrin αVβ6-positive AT2-lineage cells in vivo. Thus, we determined that these integrin αVβ6-positive cells were in the PATS-like cell state. We then assessed whether PATS-like cells also appear in BLM-treated alveolar organoids and play a role in the profibrotic function. Alveolar organoids derived from *Sftpc^CreERT2*; *Rosa26^mTmG* mice showed increased expression of PATS- and TGF-β-related genes following BLM treatment, implying that BLM-treated AT2 cells became PATS-like cells (Fig. 8b). In contrast, p53-null alveolar organoids did not show increased expression of these genes, which is consistent with a previous report that p53-cKO within AT2-lineage cells

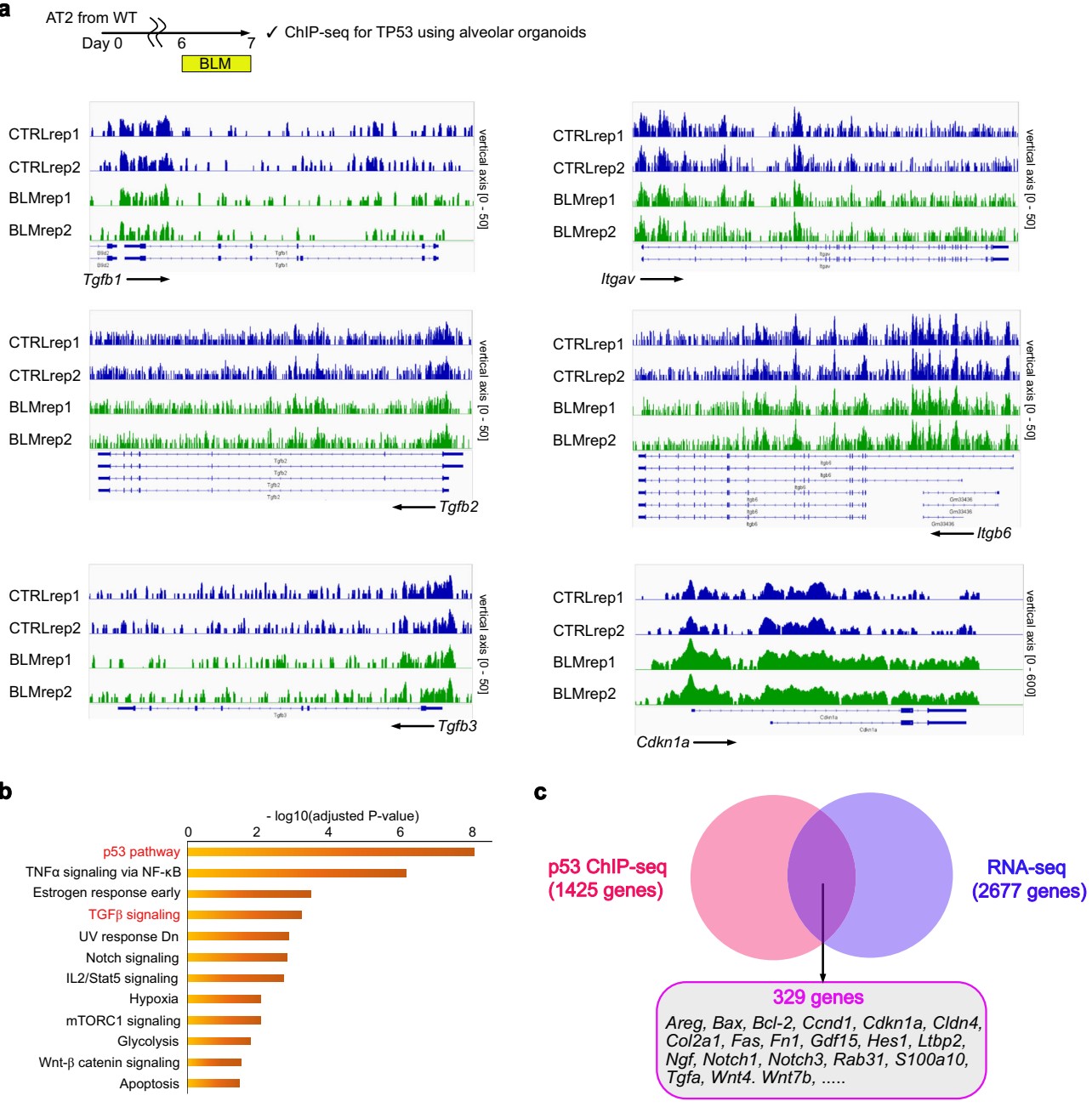

**Fig. 6 | ChIP-seq of BLM-treated alveolar organoids suggests that p53 indirectly induces TGF-β signaling. a** p53 bindings at TGF-β related genes and *Cdkn1a* (a positive control). **b** Enriched pathways of genes targeted by p53 in BLM-treated alveolar organoids (AOs) analyzed by "MSigDB Hallmark 2020". A total of 1425 genes that showed significantly higher peaks in BLM-AOs (compared to CTRL-AOs) were evaluated. Data were obtained from two independent wild-type mice. **c** Venn diagram of the genes with higher peaks in ChIP-seq and those with higher mRNA expressions in RNA-seq in BLM-AOs (compared to CTRL-AOs).

dysregulates the AT2-to-PATS transition during lung regeneration[20]. Additionally, Tgfbr2-null organoids also demonstrated tolerance to the transition into a PATS-like cell state following BLM treatment (Fig. 8b). These observations suggest that p53 and TGF-β signaling in AT2-lineage cells cooperate to induce the PATS-like transitional cell state following DNA damage.

Notably, the expression of profibrotic genes (*Tgfb1*, *Tgfb2*, *Tgfb3*, and *Itgb6*) was further downregulated in Tgfbr2-null organoids compared with that in p53-null organoids following BLM treatment (Supplementary Fig. 8a), implying that autocrine TGF-β signaling is more important than p53 signaling in determining the profibrotic potential. Based on these observations, we hypothesized that the cellular system of autocrine TGF-β signaling with positive feedback in AT2-lineage

cells is independent of p53 signaling. To test this hypothesis, we evaluated the activation of p53 signaling in Tgfbr2-null AT2 cells. BLM treatment into Tgfbr2-null alveolar organoids upregulated not only γH2AX but also total p53, its acetylated form, and downstream p21[Waf1/Cip1] expressions (Fig. 8c and Supplementary Fig. 8b). In vivo, the number of p21[Waf1/Cip1]-expressing AT2-lineage cells in BLM-treated Tgfbr2-cKO mice increased to a level comparable to that of control mice (Fig. 8d). Nevertheless, the expression of the major PATS marker KRT8 was decreased and that of the AT2 marker SFTPC was increased in p21[Waf1/Cip1]-expressing AT2-lineage cells in Tgfbr2-cKO mice (Supplementary Fig. 8c). These results indicate that p53 signaling activation does not convert AT2 cells into profibrotic PATS-like cells in the Tgfbr2-null background. To determine whether exogenous TGF-β is

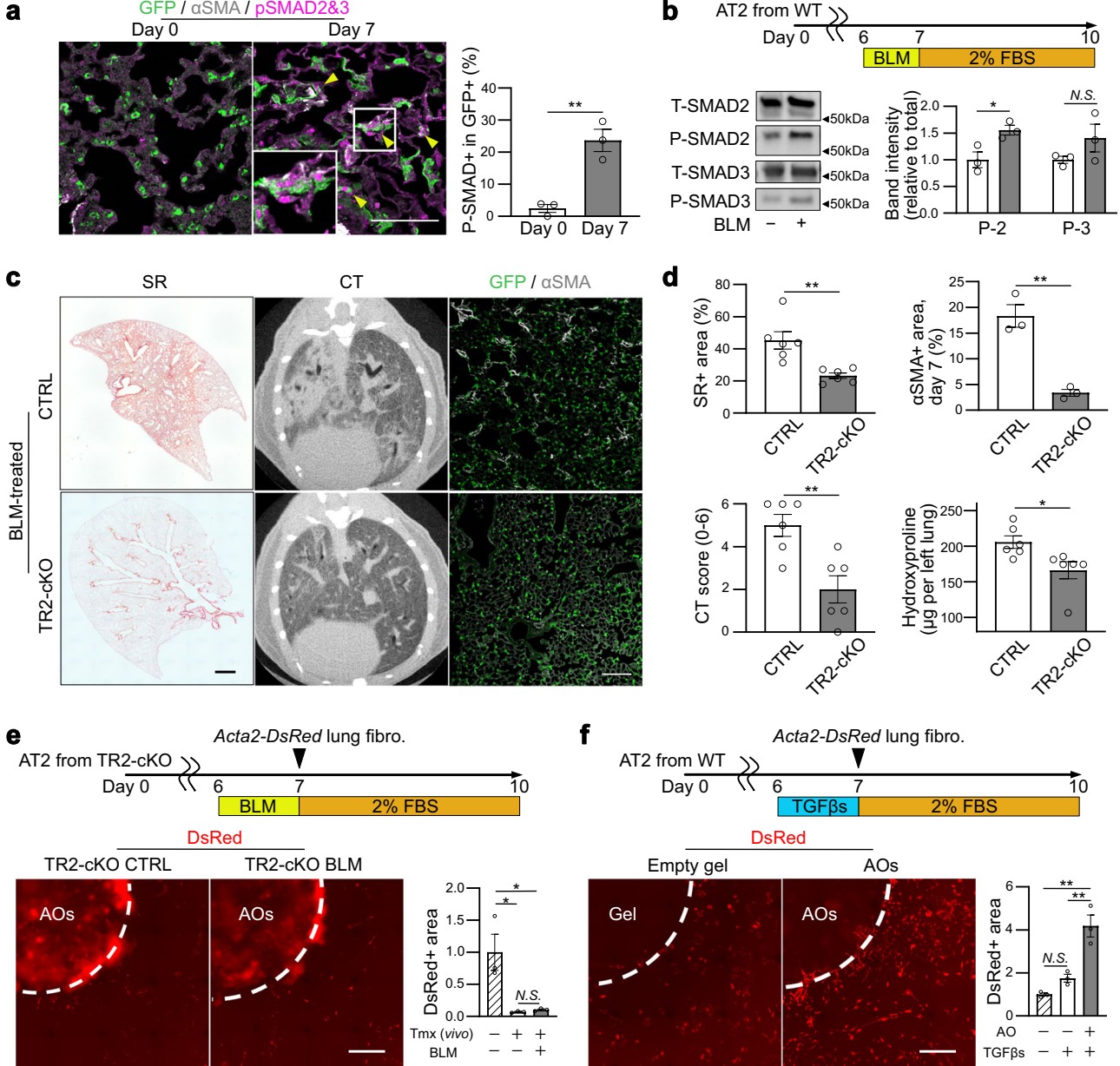

**Fig. 7 | Identification of cell-autonomous TGF-β-positive feedback loop in AT2-lineage cells. a** Representative images of immunostaining (scale bar: 100 µm) and quantification. Lung sections from BLM-treated *Sftpc^CreERT2^; Rosa26^mTmG^* mice were stained with anti-GFP, anti-phosphorylated SMAD (p-SMAD) 2&3, and anti-αSMA antibodies. Triangles indicate p-SMAD2&3⁺GFP⁺ AT2-lineage cells. Data represent the mean ± SEM of results obtained from three independent mice. **P < 0.01 compared to day 0 (unpaired, two-tailed Student's *t* test). **b** Experimental set-up, representative images of protein expression of AOs via western blotting for total/phosphorylated SMAD2/3, and quantification. The samples were derived from the same experiment and gels/blots were processed in parallel. Data represent the mean ± SEM of results obtained from three independent mice. *P < 0.05 (unpaired, two-tailed Student's *t* test). The lanes were run on the same gel. **c** Representative images of the lung sections for SR staining (scale bar: 1 mm), lung mCT, and lung sections stained with anti-GFP and anti-αSMA antibodies (scale bar: 100 µm). The *Sftpc^CreERT2^; Rosa26^mTmG^* mice (CTRL) and *Sftpc^CreERT2^; Tgfbr2^flox/flox^; Rosa26^mTmG^* mice

(TR2-cKO) were treated with BLM in vivo. **d** Quantification of SR staining (day 21, *n* = 6), mCT (day 21, *n* = 6), αSMA⁺ area (day 7, *n* = 3), and hydroxyproline in the left lungs (day 21, *n* = 6). Data represent the mean ± SEM of results obtained from three or six independent mice. *P < 0.05, **P < 0.01 (unpaired, two-tailed Student's *t*-test). **e** Representative images of co-cultured (myo)fibroblasts using TR2-cKO AO treated with BLM (100 µM, 24 h) (scale bar: 1 mm) and quantification of the relative DsRed⁺ area around the AO-containing gel. Data represent the mean ± SEM of results obtained from three independent mice. *P < 0.05 (one-way analysis of variance with Bonferroni correction). **f** Representative images of co-cultured (myo)fibroblasts using normal AO treated with mixed TGF-β1/β2/β3 (5 pg mL⁻¹ each, 24 h) (scale bar: 1 mm) and quantification of the relative DsRed⁺ area around the empty or AO-containing gel. Data represent the mean ± SEM of results obtained from three independent mice. **P < 0.01 (one-way analysis of variance with Bonferroni correction).

sufficient to make AT2 cells profibrotic even without p53 signaling, p53-null alveolar organoids were incubated with recombinant TGF-βs, and the expression of TGF-β-related genes was examined at culture days 7 and 9. Except for *Tgfb2*, these genes were markedly upregulated by day 9, and *Tgfb1*, *Itgav*, and *Itgb6* showed rapid responses by day 7

(Fig. 8e). In addition, most PATS-related genes, such as *Cldn4* and *Krt8*, were upregulated in TGF-β-treated p53-null alveolar organoids (Supplementary Fig. 8d). Therefore, TGF-β signaling in AT2 cells is sufficient to promote the acquisition of the PATS-like cell state with profibrotic potential via the positive-feedback loop. We determined that

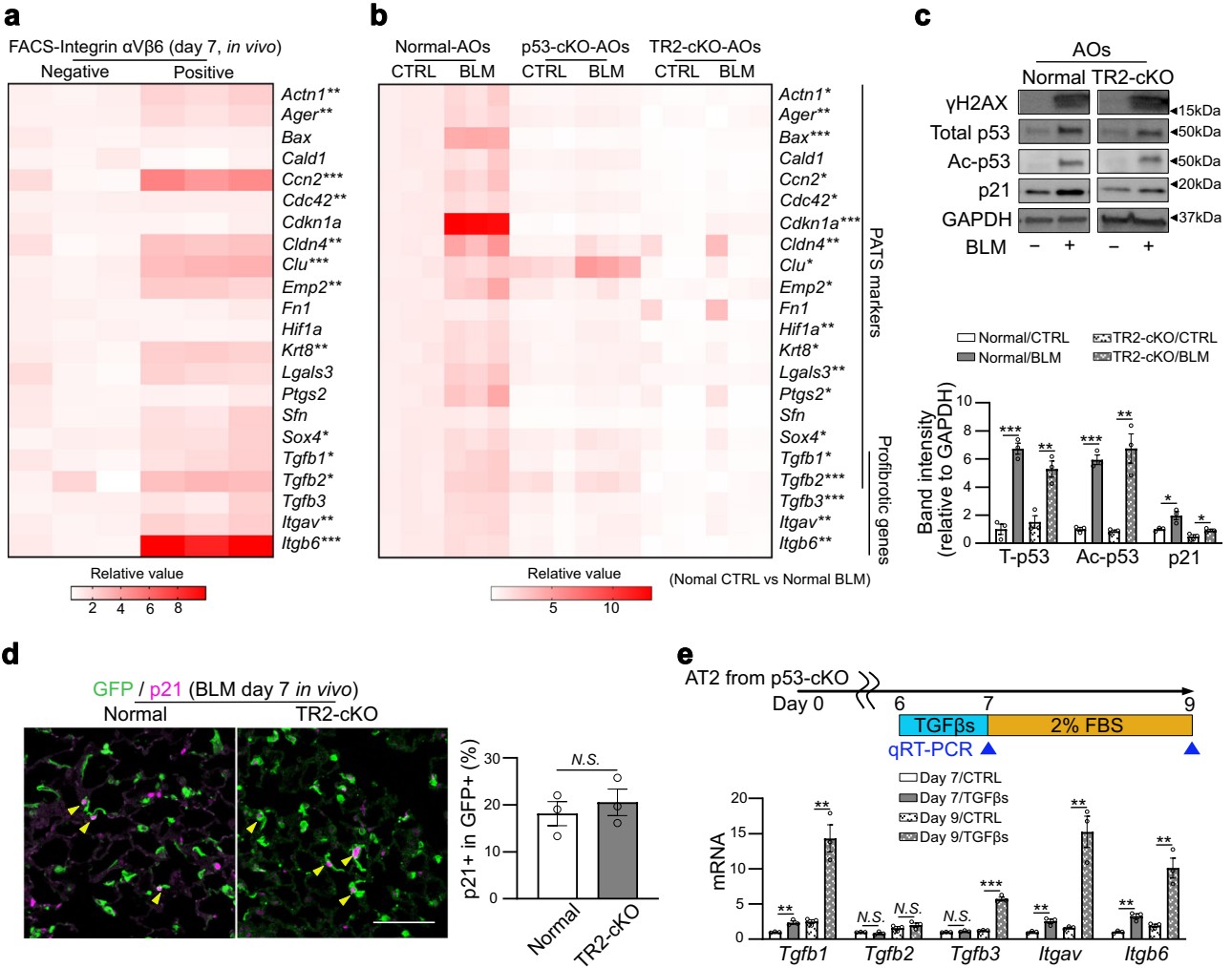

**Fig. 8 | TGF-β signaling activation drives AT2-lineage cells to transit PATS-like transitional state and to acquire profibrotic characters. a** Heatmap visualization of mRNA expression levels (for PATS-related and profibrotic genes) was performed by qRT-PCR using integrin αVβ6-positive or -negative GFP⁺ AT2-lineage cells isolated from BLM-treated *Sftpc^CreERT2; Rosa26^mTmG* mouse lungs (day 7, in vivo). Data were obtained from three independent mice. *P < 0.05, **P < 0.01, ***P < 0.001 (unpaired, two-tailed Student's t test). **b** Heatmap visualization of mRNA expression levels (for PATS-related and profibrotic genes) was performed by qRT-PCR using normal AOs (from *Sftpc^CreERT2; Rosa26^mTmG* mice), p53-cKO AOs (from *Sftpc^CreERT2; TrpS3^flox/flox; Rosa26^mTmG* mice), and TR2-cKO AOs (from *Sftpc^CreERT2; Tgfbr2^flox/flox; Rosa26^mTmG* mice). AOs were treated with BLM (100 μM, 24 h), and sampling was performed on culture day 9. Data were obtained from three independent mice. *P < 0.05, **P < 0.01, ***P < 0.001 compared between untreated CTRL normal AOs and BLM-treated normal AOs (unpaired, two-tailed Student's t test). **c** Representative images of protein expression of TR2-cKO AOs treated with BLM (100 μM, 24 h) via western blotting for γH2AX, total p53, acetylated p53 (Lys379), p21, and GAPDH. The lanes were run in the same gel but were noncontiguous. The

samples were derived from the same experiment and gels/blots were processed in parallel. The lower graph is the quantification. Data represent the mean ± SEM of results obtained from three independent mice analyzed using unpaired two-tailed Student's t test (compared to each control). **d** Representative images of immunostaining (scale bar: 100 μm) and quantification. Lung sections from BLM-treated *Sftpc^CreERT2; Rosa26^mTmG* mice (CTRL) and *Sftpc^CreERT2; Tgfbr2^flox/flox; Rosa26^mTmG* mice (TR2-cKO) were stained with anti-GFP and anti-p21 antibodies, respectively. Yellow triangles indicate p21⁺GFP⁺ AT2-lineage cells. Data represent the mean ± SEM of results obtained from three independent mice analyzed using unpaired two-tailed Student's t test. **e** Comparison of mRNA expression levels of TGF-β-related genes evaluated by qRT-PCR using p53-cKO AOs (from *Sftpc^CreERT2; Trp53^flox/flox; Rosa26^mTmG* mice). AOs were treated with mixed TGF-β1/β2/β3 (5 ng·mL⁻¹ each, 24 h), and sampling was performed on culture days 7 and 9. Data represent the mean ± SEM of results obtained from three independent mice. **P < 0.01, ***P < 0.001 compared between untreated CTRL-AOs and TGF-β-treated AOs in each time point (unpaired, two-tailed Student's t test).

p53 signaling is required to activate TGF-β-related gene expression as a trigger; however, it does not contribute to the subsequent autocrine TGF-β signaling loop in AT2(-lineage) cells.

## Profibrotic TGF-β-positive feedback loop is evolutionally conserved in human AT2 cells

Lastly, we developed a human alveolar organoid culture for a lung fibrosis model to validate that p53/TGF-β signaling-active AT2 cells directly induce fibroblast-to-myofibroblast differentiation, even in humans. We collected normal lung tissue from three patients who underwent partial pulmonary resection for suspected early-stage lung

cancer with no ILD or emphysema on their presurgical chest CT images. HTII-280⁺ human AT2 cells were sorted and cultured to create alveolar organoids (Fig. 9a). Similar to mouse alveolar organoids, BLM-treated human alveolar organoids showed obvious accumulation of γH2AX, p53, and p21^WAF1/CIF1 in the nuclei (Fig. 9b) and successfully activated the fluorescent reporter of *Acta2-DsRed* mouse lung fibroblasts (Fig. 9c), suggesting that human alveolar organoids became p53-active and profibrotic following BLM treatment. Reflecting this profibrotic effect, *TGFB1* and *ITGAV* were upregulated in BLM-treated organoids (Fig. 9d). We further analyzed the transcriptome data of lung epithelial cells from patients with IPF deposited by a previous

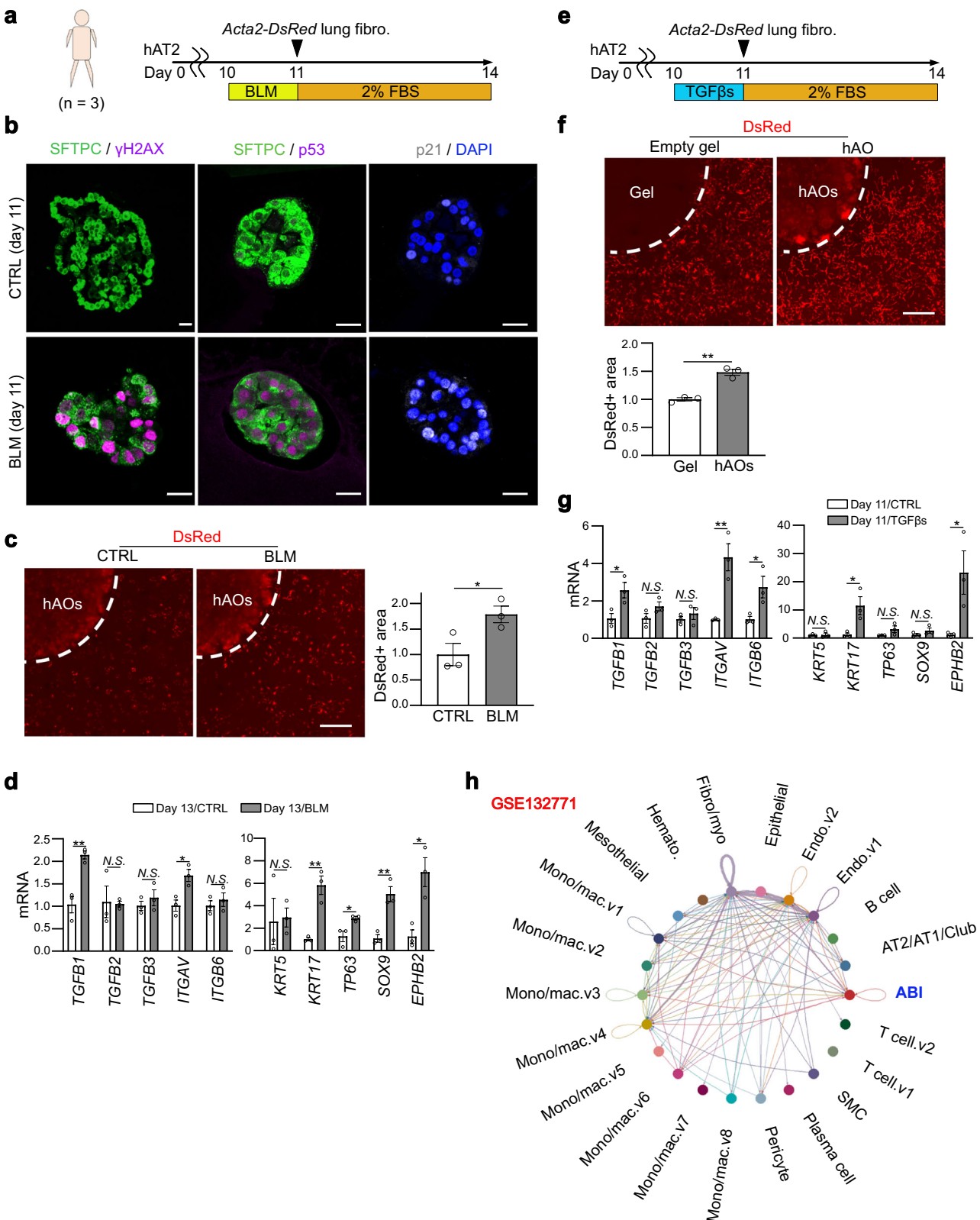

publication[38], and confirmed p53 signaling enrichment and TGF-β-related gene upregulation (Supplementary Fig. 9a, b). In human lung fibrosis and in vitro culture of human AT2 cells, alveolar-basal intermediates (ABI) (hereafter ABI cells) have been reported as a functionally homologous cell type of PATS-like cells owing to their transcriptome similarity[37,39–41]. We evaluated the markers of ABI-related genes and found that BLM-treated human alveolar organoids showed

upregulation of *KRT17*, *TP63*, *SOX9*, and *EPHB2* (Fig. 9d), suggesting the induction of an ABI-like cell state following BLM treatment in AT2-lineage cells. Furthermore, consistent with mouse organoids, incubation with recombinant TGF-βs was sufficient to induce profibrotic human alveolar organoids and to increase fibroblast-to-myofibroblast differentiation (Fig. 9e, f). TGF-β treatment upregulated TGF-β-related genes, *TGFB1*, *ITGAV*, and *ITGB6*, and ABI-related genes, *KRT17* and

**Fig. 9 | Profibrotic TGF-β-positive feedback loop is evolutionarily conserved in human AT2 cells. a** Diagram of AO culture for the lung fibrosis model using human AT2 cells and primary lung fibroblasts from *Acta2-DsRed* mice. Human lung samples were collected from three subjects. **b** Representative images of sections of hAOs treated with BLM (100 μM, 24 h) on day 11 of culture stained with anti-SFTPC, anti-H2AX, anti-p53, and anti-p21 antibodies, and DAPI (scale bar: 20 μm). For p53 and p21 expressions, an identical sphere is stained. **c** Representative images of co-cultured (myo)fibroblasts treated with BLM (100 μM, 24 h) (scale bar: 1 mm) and quantification of the relative DsRed⁺ area around the hAO-containing gel. Data represent the mean ± SEM of results obtained from three subjects. *P < 0.05 (unpaired, two-tailed Student's t-test). **d** Comparison of mRNA expression levels evaluated by qRT-PCR in BLM-treated hAOs. Data represent the mean ± SEM of results obtained from three subjects. *P < 0.05, **P < 0.01 (unpaired, two-tailed

Student's *t* test). **e** Diagram of AO culture for the lung fibrosis model using human AT2 cells and primary lung fibroblasts from *Acta2-DsRed* mice. The AOs were treated with a mixture of TGF-β1/β2/β3 (5 ng mL⁻¹ each, 24 h). **f** Representative images of co-cultured (myo)fibroblasts using hAOs treated with mixed TGF-β1/β2/β3 (scale bar: 1 mm) and quantification of the relative DsRed⁺ area around the hAO-containing gel. Data represent the mean ± SEM of results obtained from three subjects. **P < 0.01 (unpaired, two-tailed Student's *t* test). **g** Comparison of mRNA expression levels evaluated by qRT-PCR in TGF-β-treated hAOs. Data represent the mean ± SEM of the results obtained from three subjects. *P < 0.05, **P < 0.01 (unpaired, two-tailed Student's *t* test). **h** Reanalysis of single-cell RNA-seq data from lung samples isolated from patients with idiopathic pulmonary fibrosis. The interaction of TGF-β signaling between each cell type, including autocrine, was estimated using the CellChat algorithm.

*EPHB2* (Fig. 9g). However, these organoids were negative for p21[WAF1/CIP1] staining (Supplementary Fig. 9c), suggesting that TGF-β treatment induced the ABI-like profibrotic cell state in human AT2 cells without p53 signaling activation. Thereafter, we reanalyzed the dataset of single-cell RNA-seq in human IPF lungs[25] using CellChat algorism to infer cell–cell communication networks based on receptor-ligand interactions[42]. Consistent with our results, lung ABI cells in IPF lungs can be affected by TGF-β signaling in an autocrine manner (Fig. 9h and Supplementary Fig. 9d). These observations suggest an evolutionarily conserved profibrotic function of the TGF-β feedback loop in AT2 cells of mouse and human lung fibrosis.

## Discussion

In this study, we aimed to determine the mechanism of non-inflammatory lung fibrogenesis by establishing an ex vivo lung tissue culture assay and an in vitro alveolar organoid model for lung fibrosis to recapitulate fibrogenesis in vitro and exclude the involvement of immune cells. In the alveolar organoid model, we reproduced the epithelial–mesenchymal interaction with the minimum components, AT2 cells and lung fibroblasts, and successfully visualized the epithelial damage-driven fibroblast-to-myofibroblast differentiation process. Our method using primary AT2 cells, which are tissue stem cells of the alveolar epithelium, would be an alternative to other in vitro fibrosis models using pluripotent stem cell-derived AT2 cells[43,44] and is superior because of its easy approach for in vivo validation with mouse genetics. We demonstrated that DNA damage by BLM activates p53 signaling in AT2 cells to induce TGF-β-related profibrotic gene expression, which initiates a positive-feedback loop for TGF-β signaling within AT2 cells. We propose that the autocrine TGF-β signaling loop is a core cellular system that amplifies the profibrotic character of AT2 cells in non-inflammatory lung fibrogenesis, changing the cell state into PATS-like in mice or ABI-like in humans to enhance the secretion of TGF-β ligands and integrin αVβ6 (Fig. 10).

Although anti-inflammatory treatments are ineffective in human IPF[9,45], the pathogenic involvement of inflammation remains controversial. Several lines of evidence for circulating monocyte-derived alveolar macrophages have been reported in mice. Genetic or pharmacological depletion of circulating monocyte-derived alveolar macrophages can suppress BLM-induced pulmonary fibrosis until day 21[46,47]. Additionally, several studies using the BLM-induced mouse lung fibrosis model suggested that lung macrophages/monocytes can be an origin of TGFβ1 ligand[48,49]. However, in human IPF lungs, bulk RNA-seq data of tissue-resident and monocyte-derived alveolar macrophages showed no obvious upregulation of all isoforms of TGFβ[38], while the expression of *SPP1* (a gene of osteopontin) was increased, and that could enhance the "TGF-β1-induced" fibroblast-to-myofibroblast differentiation[50,51]. This is supported by the fact that the macrophage depletion could also decrease the severity of the "TGF-β1-induced" lung fibrosis model using an adenoviral vector, in which TGF-β1 ligands are already supplied from infected cells without BLM-induced epithelial DNA damage[46,47]. These previous studies suggest that the major

role of monocytes/macrophages in lung fibrosis is to enhance the effect of TGF-β1 ligands in the extracellular space, rather than to produce them. Thus, an initial supply of TGF-β ligands from AT2-lineage cells is an important initiator for lung fibrogenesis, which is also supported by a previous study reporting that the artificial selective expression of TGF-β1 at the alveolar epithelium in ex vivo lung specimens was sufficient to induce IPF-like fibrotic lung pathology[52]. Additionally, we expect that these damaged AT2-lineage cells may induce an inflammatory microenvironment that can further progress the subsequent fibrosis process. This idea is supported by our two findings:1) the timing of AT2 injury preceded that of inflammation (Fig. 1), and 2) RNA-seq and SASP analysis of BLM-treated alveolar organoids showed enrichment of inflammatory signaling, such as interferons, TNFα, and IL-6 (Figs. 3a, 5a).

Cellular senescence of AT2-lineage cells via p53 signaling is involved in lung fibrosis[15,20]; however, there is no direct evidence showing that p53-active AT2 cells directly induce myofibroblast differentiation. In this study, we provided direct evidence that p53-active AT2 cells can activate TGF-β signaling in lung fibroblasts to induce their myofibroblast differentiation. Using Tgfbr2-null alveolar organoids, we showed that AT2 cells possess a positive-feedback system of TGF-β signaling that is indispensable for the expression of TGF-β-related factors under autocrine TGF-β signaling. Notably, exogenous TGF-β was sufficient for inducing profibrotic AT2 cells, even in a p53-null background. This result suggests that p53 signaling is the initiator of TGF-β signaling. Once the TGF-β-positive-feedback loop is activated, the profibrotic characteristic is accelerated without p53 signaling. Consistent with this, even in the stretch-induced lung fibrosis model, AT2-specific knockout of *Tgfbr2* showed a marked anti-fibrotic effect, despite the enrichment of p53 signaling in the AT2-lineage cells (Supplementary Fig. 10a)[19]. Here, we propose that the TGF-β-positive-feedback loop in AT2-lineage cells is a fundamental and critical cellular system in the development of inflammation-independent lung fibrogenesis.

Clinically, pirfenidone and nintedanib are the only options for IPF pharmaceutical treatment; however, the target of these drugs is lung fibroblasts and not epithelial cells[53]. We treated our BLM-treated alveolar organoids with these two drugs and found that they did not suppress myofibroblast differentiation (Supplementary Fig. 10b). As expected, corticosteroid treatment was also ineffective in the co-culture assay. According to our findings, the TGF-β-positive feedback loop in AT2-lineage cells is an important target as a new therapeutic approach for non-inflammatory lung fibrogenesis. However, TGF-β1 knockout mice are known to exhibit systemic inflammation[54], and systemic administration of an integrin αVβ6-neutralizing antibody was ineffective in a recent clinical trial for IPF[55]. In contrast, the use of neutralizing antibodies targeting only TGF-β2 and/or β3, which can be activated by integrin-independent mechanisms, suppressed mouse BLM-induced lung fibrosis without causing systemic inflammation[56]. In addition, cellular signaling cascades from TGF-β receptors to TGF-β-related gene expression/translation in AT2-lineage cells would be

**Fig. 10 | Summary of our findings. a** DNA damage by BLM activates p53 signaling in AT2 cells to upregulate TGF-β-related profibrotic gene expression, which initiates a positive-feedback loop for TGF-β signaling within AT2 cells. This process directly induces fibroblast-to-myofibroblast differentiation even without immune cells. **b** Cascade schema in each setting (genotype/stimulation): wild-type/BLM; p53-null/BLM; Tgfbr2-null/BLM; p53-null/recombinant TGF-β.

promising therapeutic targets. To apply these TGF-β signaling suppression strategies in clinical practice, organ, cell type, and isoform specificities may be key points for success.

In this study, we could not establish a co-culture system of alveolar organoids with immune cells, such as monocytes. Future studies should evaluate whether co-culturing alveolar organoids with immune cells can further promote myofibroblast formation.

## Methods

Detailed list of the materials used in this study is summarized in Supplementary Data 1 and Supplementary Data 3.

### Mouse lines

All mice (male, 8–12 week-old) were bred and housed in a specific pathogen-free mouse facility at constant temperature (18–23 °C) and humidity (40–60%) in sterilized plastic cages. A 12h-light/12h-dark

cycle was used. *Sftpc*$^{CreERT2}$ (B6.129S-*Sftpc*$^{tm1(cre/ERT2)Blh}$/J) (#028054)[57], *Rosa26*$^{mTmG}$ (B6.129(Cg)-*Gt(ROSA)26Sor*$^{tm4(ACTB-tdTomato,-EGFP)Luo}$/J) (#007676)[58], *Tgfbr2*$^{flox/flox}$ (B6;129-*Tgfbr2*$^{tm1Karl}$/J) (#012603)[59], *Acta2-DsRed* (C.FVB-Tg(*Acta2-DsRed*)1Rkl/J) (#031159)[60], and *Sirt1*$^{flox/flox}$ (B6.129-*Sirt1*$^{tm3Fwa}$/DsinJ) (#029603)[61] mice were purchased from The Jackson Laboratory. *Pdgfra*$^{creERT2}$ mice were kindly gifted by Dr. Brigid Hogan and Dr. Christina E. Barkauskas[62]. *Trp53*$^{flox/flox}$ (C57BL/6N-Trp53<em1Rbrc>) (#RBRC09921) mice were provided from RIKEN-BRC. *Sirt1* conditional overexpression mice (*Rosa26*$^{CAG-LSL-Sirt1-P2A-eGFP}$) (Accession No. CDB0120E: https://large.riken.jp/distribution/mutant-list.html) were newly established with CRISPR/Cas9-mediated genome editing in C57BL/6 mice zygotes at RIKEN-BDR as described previously[63]. All mice were maintained on a C57BL/6 background. To label lineage cells, mice were injected with Tamoxifen (0.25 mg per g body weight) in peanuts oil 3 times on alternative days and these mice were used for all experiments three weeks after the last injection.

## Mouse phenotyping and cell preparation

Detailed protocols and procedures for the following methods are summarized in Supplementary Information: Induction of lung fibrosis by intratracheal administration of bleomycin; Assessment of lung mCT; Bronchoalveolar lavage (BAL) of mouse lungs; Preparation of mouse primary cells and FACS sorting; Preparation of human primary AT2 cells; and MACS sorting.

## Culture of alveolar organoids

Sorted AT2 cells (GFP$^+$ cells from *Sftpc$^{CreERT2}$: Rosa26$^{mTmG}$* mouse lung cells or EPCAM$^+$MHCII$^+$CD31$^-$CD45$^-$ from wild-type mouse lung cells[64]; HTII-280$^+$ in humans[65]) were mixed in culture medium and the equal volume of growth factor-reduced Matrigel (Corning) and 20000 cells per 20 μL gel-containing drops were placed on the bottom of plates. The drops were incubated for 20 min at 37 °C to solidify the gels, and 400 μL or 2 mL of medium was added to each well in 24-well or 6-well plates, respectively. The composition of culture media is described in Supplementary Data 1[66]. Medium was changed every 3 days. Y27632 (10 μM) was included in the medium for the first 3 days of the sorting or passaging to increase cell viability. On day 7 in mice and day 14 in humans, a passage was performed (mouse: 5000 cells per 20 μL gel-containing drop; human: 10000 cells per 20 μL gel-containing drop) and used for experiments from day 6 in mice and day 10 in humans. Before administration of specific drugs or growth factors, washing using DPBS (3 min at room temperature) was performed twice. BLM was used at a concentration of 100 μM. Organoids were imaged using IX83 fluorescent microscope (Olympus). To evaluate organoid sizes, two organoid-containing drops per mouse were quantified. The size of the top 20 spheres in each drop was measured, which was repeated for three independent mice.

## Single-/co-culture of mouse fibroblasts

Sorted primary fibroblasts (PDGFRα$^+$EPCAM$^-$CD31$^-$CD45$^-$CD146$^-$LYVE1$^-$) from mouse lungs were cultured[67]. For single culture in 96-well plates, 2000 cells in 200 μL of a basic medium containing 2% FBS with or without specific drugs, growth factors, or chemokines were used in each well. For co-culture with mouse/human alveolar organoids or empty gel in 24-well plates, 15,000 cells in 350 μL basic medium or equal dose of organoid culture supernatant (both with 2% FBS) were used in each well. Organoid supernatant was collected after centrifugation (1000 g, 10 min). On day 3 of (co-)culture, DsRed fluorescence of the (myo)fibroblasts was imaged with/without staining of Hoechst 33342 (Dojindo). For the live imaging, CellDiscoverer 7 (Carl Zeiss) was used.

## Ex vivo culture of mouse lung lobes

Mice were sacrificed by carbon dioxide and perfused with 5 mL of a saline solution through the right ventricle. After three times of BAL using 0.8 mL sterile PBS, 0.8 mL of BLM (700 μM) or PBS (as a control) was intratracheally injected and the trachea was tied with a string. Then the lobes were incubated for 30 min and the peripheral area was cut into about 5 mm × 5 mm size. The specimens were cultured in a basic medium containing 2% FBS with/without BLM (100 μM) for 24 h. On the next day, the medium was changed into a fresh basic medium containing 2% FBS after washing it with DPBS twice. Sampling was done on days 2 and 12.

## Other methods

Detailed protocols and procedures for the following methods are summarized in Supplementary Information: Bulk RNA-seq of mouse alveolar organoids; Total RNA isolation, cDNA preparation, and quantitative RT-PCR; Preparation of lung or organoid sections; Collagen staining of lung sections and the quantification; Immunohistochemistry; Staining of senescence-associated β-galactosidase; Proximity ligation in situ hybridization (PLISH); Hydroxyproline assay; Western blotting; ChIP-seq; and Reanalysis of datasets of single-cell RNA-seq.

## Statistics and reproducibility

All results are presented as mean ± standard error from a minimum of three independent experiments. Statistical analyses were performed using unpaired Student's *t* test (two-tailed) for comparisons between two groups or using a one-way analysis of variance with Bonferroni correction for comparisons between more than two groups. The exact *P* values are provided in Source Data file and differences with *P* < 0.05 are considered significant. Experiments for section staining were repeated independently with similar results at least three times. The representative micrographs are presented in each figure.

## Study approval

We handled the mice in accordance with the ethics guidelines of the institute, and all the experimental procedures using animals have been approved by the Institutional Animal Care and Use Committee of RIKEN Kobe Branch (K2020-020). The use of human lung samples has been also approved by the Institutional Review Board of Kobe University Graduate School of Medicine (B2056707). Informed consent has been obtained from all the patients.

## Reporting summary

Further information on research design is available in the Nature Portfolio Reporting Summary linked to this article.

## Data availability

The authors declare that all data supporting the findings of this study are available within the article and its supplementary materials, including Source Data. The raw images are available in the System Science of Biological Dynamics (SSBD) repository (URL: https://ssbd.riken.jp/repository/288/; https://doi.org/10.24631/ssbd.repos.2023.04.288). Our original sequencing data in this study are available at NCBI GEO under the accession numbers GSE211531 (RNA-seq) and GSE231445 (ChIP-seq). Previously published sequencing data that were reanalyzed are available under the accession number GSE138585[19], GSE132910[15], and GSE132771[25]. Source data are provided with this paper.

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

## Acknowledgements

We thank Dr. Brigid Hogan for kindly providing *Pdgfra*<sup>CreERT2</sup> mice, Dr. Hiroshi Mizuma for his technical support during mCT imaging, and RIKEN-BRC for supplying *Trp53*<sup>flox/flox</sup> mice. Regarding RNA-seq analysis, we also thank Laboratory for Phyloinformatics at RIKEN-BDR and the members: Dr. Kaori Tatsumi (for library preparation and quality check) and Dr. Shigehiro Kuraku (for on-site supervision). We also thank Dr. Shogo Nakayama for his advice on reanalyzing the scRNA-seq dataset.

## Author contributions

Y.E. and M.M. conceived the project. Y.E., A.O., S.B., and A.Y. conducted experiments and processed data. Y.E., H.K., T.F., and M.M. analyzed data and interpreted results. Y.E., O.N., and Mi.Ka. conducted bulk RNA-seq and ChIP-seq using alveolar organoids and analyzed the data. D.H., Y.T., Y.M., and T.N. obtained human lung specimens. T.F., Mi.Ki., and T.A. generated Sirt1-OE mice. Y.E. and M.M. wrote the original manuscript. All authors reviewed the manuscript and approved the submission.

## Funding

This manuscript received support from the Japan Society for the Promotion of Science for Early-Career Scientists (20K17238) (Y.E.); for Challenging Exploratory Research (19K22630) (M.M.); Okamoto Foundation for lung fibrosis (2019) (Y.E.); G-7 Foundation (2020) (M.M. and Y.E.); and Special Postdoctoral Researcher Program of RIKEN (Y.E.).

## Competing interests

T.F. is an employee of Otsuka Pharmaceutical Co., Ltd, but there is no financial competing interest in this study. The other authors declare no competing interests.
