## [Peer Review File · Nature Communications]

REVIEWER COMMENTS

Reviewer #1 (Remarks to the Author):

The manuscript by Enomoto et al. establishes an alternative methodology by which to assess how epithelial and mesenchymal cells contribute to a lung fibrosis phenotype in the absence of immune cells. They accomplish this by using ex vivo and in vitro models with AT2 cells exposed to bleomycin with and without other genetic modifications and examine fibroblast to myofibroblast transition as well as other markers of fibrosis. In this work, they propose an interesting paradigm by which bleomycin induces epithelial cell double-stranded DNA damage that results in a “senescent AT2” or PATS-like transitional identity with p53 and p21 activation that then leads to initiation of a TGF β positive feedback loop in the AT2 cells to promote a fibrotic phenotype. This study is a tremendous amount of work including bioinformatics as well as multiple ex vivo and in vitro studies utilizing their Acta2-DsRed reporter model to arrive at the proposed mechanism in supplemental figure 8. I think that this work is suitable for publication in Nature Communications with revisions.

Major Critiques

1. There are no figures with co-staining of an AT2 specific marker with their GFP to definitively show that they are indeed AT2 cells and not AT1 cells derived from the AT2 lineage. Specifically, in figures 6A as well as 7D, the cells marked with arrows denoting pSMAD2/3 or p21 staining have AT1 cell morphology or perhaps are elongated AT2s consistent with a PATS-like morphology. The authors should perform co-staining for At1 and AT2 marker along with the lineage trace to reveal these data. The quantifications of %GFP+/marker of interest should only include GFP+AT2 cells. The in vitro feeder-free AT2 model also appears to contain AT1 cells as shown in Figure 1F. AT2-AT1 reprogramming following bleomycin is well-described, so if the AT1 cells are also potentially contributing to the phenotype, that should be stated.
2. In Figure 1, the authors state that the sentinel event following bleomycin is epithelial cell double-stranded DNA damage. However, in the IHC staining on Figure 1B, in the damaged area, γ H2AX is present throughout in cells other than the AT2. The authors should (A) quantify % AT2 cell + γ H2AX/ total AT2 in the damaged area and (B) perform co-staining for AT1 cells and perhaps endothelial cells to see if epithelial cells are indeed the predominantly damaged cell type.
3. Although TGF β is known to affect cell fate, it may also affect cell differentiation, proliferation, and apoptosis. In addition, Bax, a pro-apoptotic gene was significantly upregulated in their BLM-AOs as shown in Figure 3B. The authors should evaluate for apoptosis and proliferation in their AT2 organoids. In the figures containing the Dsred reporter system, with the presented data the authors cannot definitely say that the increase in Dsred area is strictly secondary to fibroblast to myofibroblast transition and not proliferation. Although the end-result of a pro-fibrotic phenotype and increased myofibroblast number would not change, this information would still be important to further define the underlying mechanism. Specifically, I would recommend that the authors assess for proliferation of fibroblasts/myofibroblasts in Figure 3F and 6F.

Minor Critiques

1. The protective effect of Tgfr2 loss following bleomycin administration is already known although a different promoter (Nkx2.1) is used. The authors should include this paper in their manuscript. Li M, Krishnaveni MS, Li C, Zhou B, Xing Y, Banfalvi A, Li A, Lombardi V, Akbari O, Borok Z, Minoo P. Epithelium-specific deletion of TGF- β receptor type II protects mice from bleomycin-induced pulmonary fibrosis. J Clin Invest. 2011 Jan;121(1):277-87.
2. The authors should provide a simplified schematic describing their findings in their main figures.
3. The AO staining in Figure 8B for Sftpb and H2AX co-localization could be improved.
4. The wording in the abstract on line 17 regarding human organoids should be revised.
5. Although the end of the abstract highlights the autocrine AT2 positive feedback loop, it was not mentioned earlier when describing the data. The authors should mention it

Reviewer #2 (Remarks to the Author):

In the manuscript "Autocrine TGF- β -positive feedback in profibrotic AT2-lineage cells plays a crucial role in non-inflammatory lung fibrogenesis" by Enomoto et al., the authors use multiple mouse and organoid models to study the interaction between AT2-lineage cells and fibroblasts. The authors conclude that autocrine TGF-beta signalling within profibrotic AT2-lineage cells primes them to induce fibroblast-to-myofibroblast differentiation.

Using ex vivo culture of lung lobe pieces and alveolar organoids from Acta2-DsRed and Sftpc-mTmG mice they show that ex vivo bleomycin treatment induces a DNA damage together with p53 activation in AT2 cells that causes a transcriptomic shift towards previously described PATS/Krt8+ ADI cells. The profibrogenic role of PATS/Krt8+ ADI cells has been proposed but to date was not clearly demonstrated experimentally. The authors show that bleomycin primed AT2-lineage cells induce Acta2 and Col1a1 expression in neighbouring fibroblasts, suggesting a fibroblast-myofibroblast switch. Using experiments with p53 null and TGFbR null mice the authors attempt to decouple the role of p53 and TGFb in this profibrogenic epithelial-mesenchymal interaction. Based on their data, the authors suggest that in agreement with previous literature p53 signaling is important to reach the PATS/Krt8+ ADI state. The authors data also suggests that the paracrine effect on fibroblasts seems dependent on autocrine TGFbeta signalling in AT2-lineage cells in mouse organoids.

Overall, this study adds important new data to the field. The differential role of p53 and TGFbeta pathways is very interesting and should be addressed more. In particular, it would be important to further comment and address how p53 activity in AT2-lineage cells is required for initial availability of active TGFbeta in early stages of fibrogenesis as discussed by the authors.

The reviewer has specific comments and suggestions below.

- What is the role of p53 in the initiation of the paracrine profibrogenic function of the AT2-lineage cells? How does it initiate the TGFbeta autocrine loop? Is this just dependent on direct induction of TGFb and Integrin α V β 6 expression?

- The authors show that activation of p53 with Nutlin-3a is sufficient for inducing the profibrotic programme in AT2-lineage cells, even in the absence of BLM. Is it also sufficient for inducing TGFbeta and integrin α V β 6? Is this also the case in human?

- Why is the AT2 specific p53 null mouse having less fibrosis if active TGFbeta could be sufficient to induce profibrotic loop? The authors suggest a critical and exclusive role of AT2-lineage for TGF-beta activation in early stages of fibrogenesis. Do the authors propose that ALL active TGFbeta initially depends on the p53-activated alveolar epithelium? If this depends on p53 pathway activity, can they link this to a potential molecular mechanism?

- Was the TGFbeta treatment of organoids for conditioned media swap done on organoids without fibroblasts? The only evidence for TGFbeta being sufficient to bypass p53 is conditioned media swap experiments. In these experiments organoids, which have both epithelial and mesenchymal cells) were treated with TGFbeta and conditioned media was transferred to reporter fibs. What if exogenous TGFbeta activated fibroblast in organoids and this contributed to the CM effect because there is also an unseen autocrine loop in the fibroblasts?

- The authors hypothesize that p53-driven fibroblast-to-myofibroblast differentiation is mediated by activating latent TGF-beta through increased integrin α V β 6 expression. They tested this hypothesis by adding a neutralizing antibody against integrin α V β 6 in an AT2-fibroblast co-culture. However, the antibody will block the integrin both on AT2 and fibroblasts. Can the authors show direct induction of integrin α V β 6 by p53 activation in AT-lineage cells?

- Based on the cellchat analyses in figure 8H, the authors claim that epithelial cells can be affected by TGFb feedback loops. We believe more in-depth cell-cell communication analyses are necessary to reinforce this claim, both in terms of epithelial cell granularity as well as in the depth of analysis itself (e.g. quantification of the specific ABI-ABI TGFb signaling compared to other cell-cell TGFb signaling strength and the role of TGFBR2 herein).

- Other small comments:

o In the manuscript, the authors use a Acta2-DsRed mouse line to investigate fibroblast-to-myofibroblast differentiation. However, scRNASeq studies have shown that myofibroblast identity is mostly characterised by Cthrc1 and Tnc, rather than Acta2. Moreover, cthrc1^{high} myofibroblasts were

shown to have rather limited Acta2 expression. The authors need to show more evidence that the DsRed+ cells are indeed myofibroblasts. Similarly, the authors show Acta2 mRNA levels in figure 5E, suppl Fig 1 and suppl fig 3, which is insufficient to prove myofibroblast identity.

o The authors use western blot to show protein expression, however no quantification is provided. This should be done, including statistical analysis.

o The terms AT2-lineage cells and AT2 cells are both used in the manuscript in the context of the organoid and mouse models, leading to confusion. (This comment obviously does not apply to the section on human AT2 cells)

Reviewer #3 (Remarks to the Author):

In this manuscript the authors describe the effects of p53 and TGFbeta in eliciting non-inflammatory pulmonary fibrosis. Using genetically engineered mice, and organoid cultures as experimental systems they assess the role of damaged AT2 cells as critical mediators of fibrogenesis. Although much of the work confirms previous studies on idiopathic lung fibrosis the study does add new insights. It illustrates that no major role for immune cells has to be invoked in IPF although a further enhancing effect by factors produced by macrophages, neutrophils or other infiltrating immune cells is not excluded. They also show that the autocrine TGFbeta signaling in damaged AT2 cells plays a critical role in this process as knocking out the Tgfbr2 receptor in AT2 cells abrogates the fibrotic response in lung fibroblasts. The conclusions are supported by detailed and well-performed genetic experiments. However, there are a few issues the authors should address to make the manuscript suitable for publication in Nature Comm.

- In the first part of the manuscript the authors put a lot of emphasis on p53 signaling as a critical factor instructing AT2 cells to initiate the fibrotic process. However, other groups have performed more detailed analyses how p53 might more directly affect gene expression relevant for this process (e.g. as shown for the effects seen for PATS, Kobayashi et al. Nature Cell Biol. 2020). Therefore, this can be described more succinct with more extensive reference to recent publications describing the well published p53-TGFbeta- integrin alphaVbeta6 axis.

- Given the prominent role ascribed to p53 signaling in this study, the western blots to support the induction of p53 and Pser15-p53 (figures 3c, 4c, and their respective supplementary figures) are not very convincingly showing this. One would expect a more robust induction than is shown. Are other inducers of senescence such as INK4a/b, Mdm2, p19Arf, and p63 more convincingly induced, as senescence is a prominent feature of Bleomycin treatment AT2 cells as well as of PATS.

- The experiments illustrating a role for TGF-beta in the AT2 lung fibroblast interaction are more revealing. However, it remains unclear for the reader to what extent TGF-beta itself can induce the full fibrotic response achieved with the conditioned medium of bleomycin-treated AT2 cells. Although the authors tested a series of SASP factors produced by BLM treated AT2 cells, and conclude that TGFbeta is the only factor inducing the fibrotic response (as also depicted in the explanatory diagram) it remains unclear whether the maximal fibrotic response in lung fibroblast induced by the lowest concentration of TGFbeta can be further enhanced by the addition of conditioned medium from BLM treated AT2 cells, as this would point to additional (not tested) factors that can synergize with TGFbeta in augmenting fibrosis.

Point-by-point responses to the Reviewers' comments

Reviewer 1

The manuscript by Enomoto et al. establishes an alternative methodology by which to assess how epithelial and mesenchymal cells contribute to a lung fibrosis phenotype in the absence of immune cells. They accomplish this by using ex vivo and in vitro models with AT2 cells exposed to bleomycin with and without other genetic modifications and examine fibroblast to myofibroblast transition as well as other markers of fibrosis. In this work, they propose an interesting paradigm by which bleomycin induces epithelial cell double-stranded DNA damage that results in a "senescent AT2" or PATS-like transitional identity with p53 and p21 activation that then leads to initiation of a TGF β positive feedback loop in the AT2 cells to promote a fibrotic phenotype. This study is a tremendous amount of work including bioinformatics as well as multiple ex vivo and in vitro studies utilizing their Acta2-DsRed reporter model to arrive at the proposed mechanism in supplemental figure 8. I think that this work is suitable for publication in Nature Communications with revisions.

Major Critiques 1. There are no figures with co-staining of an AT2 specific marker with their GFP to definitively show that they are indeed AT2 cells and not AT1 cells derived from the AT2 lineage. Specifically, in figures 6A as well as 7D, the cells marked with arrows denoting pSMAD2/3 or p21 staining have AT1 cell morphology or perhaps are elongated AT2s consistent with a PATS-like morphology. The authors should perform co-staining for At1 and AT2 marker along with the lineage trace to reveal these data. The quantifications of %GFP+/marker of interest should only include GFP+AT2 cells. The in vitro feeder-free AT2 model also appears to contain AT1 cells as shown in Figure 1F. AT2-AT1 reprogramming following bleomycin is well-described, so if the AT1 cells are also potentially contributing to the phenotype, that should be stated.

Response:

We are grateful for the reviewer's suggestion. As indicated, our data lacked analysis of AT2 differentiation or cell specification. Therefore, we have added IHC analysis data for pro-SP-C (AT2)/KRT8 (PATS)/HOPX (AT1) and presented the results in Supplementary Figs. 6A and 7C. These revised figures show that TGF- β signaling promotes AT2 differentiation into PATS *in vivo*, which appears to be independent of p53 signaling activation.

Supplementary Fig. 6A

Supplementary Fig. 7C

2. In Figure 1, the authors state that the sentinel event following bleomycin is epithelial cell double-stranded DNA damage. However, in the IHC staining on Figure 1B, in the damaged area, γ H2AX is present throughout in cells other than the AT2. The authors should (A) quantify % AT2 cell + γ H2AX/ total AT2 in the damaged area and (B) perform co-staining for AT1 cells and perhaps endothelial cells to see if epithelial cells are indeed the predominantly damaged cell type.

Response:

Thank you for your suggestion. Accordingly, we have added data for co-staining of γ H2AX with pro-SP-C (AT2)/HOPX (AT1)/ERG (endothelial cells). AT2 cells frequently exhibited upregulated γ H2AX expression compared to other-type cells in BLM-treated lobes, although it is not statistically significant. Please refer to the revised Supplementary Fig.1B. To emphasize the importance of DNA damage in AT2 cells, even *in vivo*, we moved several panels from Supplementary Fig.1 to the revised main Fig.1.

Supplementary Fig. 1B

B

3. Although TGF β is known to affect cell fate, it may also affect cell differentiation, proliferation, and apoptosis. In addition, Bax, a pro-apoptotic gene was significantly upregulated in their BLM-AOs as shown in Figure 3B. The authors should evaluate for apoptosis and proliferation in their AT2 organoids.

Response:

Thank you for the suggestion. We have accordingly evaluated the apoptosis and proliferation of our organoids using size quantification, Annexin-PI assay, and Ki67 staining/mRNA analysis (Revised supplementary Fig. 3A–D). These analyses revealed that treatment with BLM promoted apoptosis and decreased the proliferation of cultured AT2 cells.

Supplementary Fig. 3

4. In the figures containing the Dsred reporter system, with the presented data the authors cannot definitely say that the increase in Dsred area is strictly secondary to fibroblast to myofibroblast transition and not

proliferation. Although the end-result of a pro-fibrotic phenotype and increased myofibroblast number would not change, this information would still be important to further define the underlying mechanism. Specifically, I would recommend that the authors assess for proliferation of fibroblasts/myofibroblasts in Figure 3F and 6F.

Response:

Thank you for your excellent suggestion. We agree that this is an important point and have accordingly assessed the proliferation of fibroblasts/myofibroblasts in our fibrosis organoid model by counting the cell number at the end of the co-culture period. In this system, the cell number of fibro/myo was not increased with the BLM-treated organoids and was slightly increased by the treatment with TGF- β . Because the increased DsRed-positive area was three times or more that of each control, we believe that the main mechanism of the area increases under these conditions would be explained by differentiation, and not proliferation, into DsRed-positive myofibroblasts.

Fig. 3F

Supplementary Fig. 6

Minor Critiques 1. The protective effect of Tgfbr2 loss following bleomycin administration is already known although a different promoter (Nkx2.1) is used. The authors should include this paper in their manuscript. Li M, Krishnaveni MS, Li C, Zhou B, Xing Y, Banfalvi A, Li A, Lombardi V, Akbari O, Borok Z, Mino P. Epithelium-specific deletion of TGF- β receptor type II protects mice from bleomycin-induced pulmonary fibrosis. J Clin Invest. 2011 Jan;121(1):277-87.

Response:

We apologize for not including this important report. We have now mentioned and cited the paper in the revised manuscript as follows (page 17, lines 6–8):

“...which was consistent with the results of ex vivo lung-lobe culture (Supplementary Fig. 6C) and a previous report using Nkx2.1-cre driver³⁴.”

2. The authors should provide a simplified schematic describing their findings in their main figures.

Response:

As the reviewer kindly suggested, we have added schematic figures in the revised main Fig. 10.

Fig. 10.

3. The AO staining in Figure 8B for Sftpb and H2AX co-localization could be improved.

Response:

Thank you for your comment. We have accordingly revised Fig. 9 and added human AO staining for SFTPC and γ H2AX.

Fig. 9B.

4. The wording in the abstract on line 17 regarding human organoids should be revised.

5. Although the end of the abstract highlights the autocrine AT2 positive feedback loop, it was not mentioned earlier when describing the data. The authors should mention it

Response:

The abstract was elaborated in response to these comments and further revised to less than 150 words to conform to the journal's guidelines.

Revised abstract: "The molecular etiology of idiopathic pulmonary fibrosis (IPF) has been extensively investigated to identify new therapeutic targets. Although anti-inflammatory treatments are not effective for patients with IPF, damaged alveolar epithelial cells play a critical role in lung fibrogenesis. Here, we established an organoid-based lung fibrosis model using mouse and human lung tissues to assess the direct communication between damaged alveolar type II (AT2)-lineage cells and lung fibroblasts by excluding immune cells. Using this *in vitro* model and mouse genetics, we demonstrated that bleomycin causes DNA damage and activates p53 signaling in AT2-lineage cells, leading to AT2-to-AT1 transition-like state with a senescence-associated secretory phenotype (SASP). Among SASP-related factors, TGF- β played an exclusive role in promoting lung fibroblast-to-myofibroblast differentiation. Moreover, the autocrine TGF- β -positive feedback loop in AT2-lineage cells is a critical cellular system in non-inflammatory lung fibrogenesis. These findings provide insights into the mechanism of IPF and potential therapeutic targets.

Reviewer 2

In the manuscript “Autocrine TGF- β -positive feedback in profibrotic AT2-lineage cells plays a crucial role in non-inflammatory lung fibrogenesis” by Enomoto et al., the authors use multiple mouse and organoid models to study the interaction between AT2-lineage cells and fibroblasts. The authors conclude that autocrine TGF-beta signalling within profibrotic AT2-lineage cells primes them to induce fibroblast-to-myofibroblast differentiation. Using ex vivo culture of lung lobe pieces and alveolar organoids from Acta2-DsRed and Sftpc-mTmG mice they show that ex vivo bleomycin treatment induces a DNA damage together with p53 activation in AT2 cells that causes a transcriptomic shift towards previously described PATS/Krt8+ ADI cells. The profibrogenic role of PATS/Krt8+ ADI cells has been proposed but to date was not clearly demonstrated experimentally. The authors show that bleomycin primed AT2-lineage cells induce Acta2 and Col1a1 expression in neighbouring fibroblasts, suggesting a fibroblast-myofibroblast switch. Using experiments with p53 null and TGFbR null mice the authors attempt to decouple the role of p53 and TGFb in this profibrogenic epithelial-mesenchymal interaction. Based on their data, the authors suggest that in agreement with previous literature p53 signaling is important to reach the PATS/Krt8+ ADI state. The authors data also suggests that the paracrine effect on fibroblasts seems dependent on autocrine TGFbeta signalling in AT2-lineage cells in mouse organoids. Overall, this study adds important new data to the field. The differential role of p53 and TGFbeta pathways is very interesting and should be addressed more. In particular, it would be important to further comment and address how p53 activity in AT2-lineage cells is required for initial availability of active TGFbeta in early stages of fibrogenesis as discussed by the authors. The reviewer has specific comments and suggestions below.

-What is the role of p53 in the initiation of the paracrine profibrogenic function of the AT2-lineage cells? How does it initiate the TGFbeta autocrine loop? Is this just dependent on direct induction of TGFb and Integrin α V β 6 expression?

Response:

Thank you for your important and valid questions. We agree that we should have focused more on the role of p53 in the initiation of the autocrine TGF- β feedback loop. In this revision, we performed ChIP-seq with an anti-p53 antibody for BLM-treated alveolar organoids (AO) to identify the direct targets of p53 in AT2 cells.

Although small peaks upstream of *Tgfb1*, *2*, *3*, and *Itgav* were observed, we could not identify apparent differences between CTRL-AO and BLM-AO. This result suggests that p53 does not directly induce the expression of these profibrotic TGF- β genes, at least in our assay. However, the direct p53 target genes identified by our ChIP-seq analysis included several TGF- β -related genes. We identified 329 genes that were the direct targets of p53 and upregulated in our RNA-seq data, including genes that have been reported to have a potential to induce or activate TGF- β signaling, such as *Areg*, *Cdkn1a*, *Ltbp2*, *Pdgfc*, and *Tgfa*. These data have been added to the revised Fig. 6 and Supplementary Table 4. These results have been described in the revised manuscript on Page 15, lines 9–18 and Page 16, lines 1–4 as follows:

“A time lag between p53 activation (day 7) and profibrotic TGF- β -related genes (day 9) prompted us to

determine whether TGF- β isoforms and integrins are direct targets of p53. Therefore, we performed ChIP-seq analysis using an anti-p53 antibody for BLM-treated alveolar organoids to determine the direct targets of p53. We observed peaks upstream p53 target genes, such as *Cdkn1a* (Fig. 6A). However, there were no apparent differences between control- and BLM-treated-alveolar organoids in the binding of p53 to TGF- β isoforms and integrins (Fig. 6A), suggesting that p53 does not directly induce these profibrotic factors. However, the direct p53 target genes identified by our ChIP-seq analysis included several TGF- β -related genes (Fig. 6B). We identified 329 genes that were the direct targets of p53 and upregulated in our RNA-seq data, including genes that have been reported to have a potential to activate TGF- β signaling, such as *Areg*²⁹, *Cdkn1a*³⁰, *Ltbp2*³¹, *Pdgfr32*, and *Tgfa*³³ (Fig. 6C and Supplementary Table 4). These data suggest that p53 indirectly upregulates profibrotic factors by inducing these candidate factors.”

Fig. 6

-The authors show that activation of p53 with Nutlin-3a is sufficient for inducing the profibrotic programme in AT2-lineage cells, even in the absence of BLM. Is it also sufficient for inducing TGFbeta and integrin α Vb6? Is this also the case in human?

Response:

Thank you for your question. In mouse organoids, Nutlin-3a was sufficient for inducing *Tgfb1* ($p < 0.01$), *Itgb6* ($p < 0.05$), and likely *Itgav* as well ($p < 0.1$). We have added the data to the revised supplementary Fig .5D.

However, in human organoids, the effect of Nutlin-3a failed to show statistical significance, probably due to the high variability of human tissues, which might result in individual differences in the sensitivity to the drug. Please see the graph below.

-Why is the AT2 specific p53 null mouse having less fibrosis if active TGFbeta could be sufficient to induce profibrotic loop?

The authors suggest a critical and exclusive role of AT2-lineage for TGF-beta activation in early stages of fibrogenesis.

Do the authors propose that ALL active TGFbeta initially depends on the p53-activated alveolar epithelium? If this depends on p53 pathway activity, can they link this to a potential molecular mechanism?

Response:

Thank you for your comment. As we have shown in supplementary Fig 5C, the expressions of *Tgfb1*, *Tgfb3*, *Itgav*, and *Itgb6* of BLM-treated organoids were significantly lower in p53-cKO case. This result suggests that p53 signaling plays a dominant role in the induction of the TGF- β signaling loop, which would explain why AT2-specific p53 null mice showed less fibrosis. This concept could be supported by the results of a recent paper (Yao C, et al. AJRCCM 2021). The re-analyzed data have been added to the revised supplementary Fig. 5E (as shown below). Regarding the molecular mechanism, as we explained above, we performed ChIP-seq with an anti-p53 antibody for BLM-treated alveolar organoids (AO) to identify the direct targets of p53 in AT2 cells.

Although small peaks upstream of *Tgfb1*, *2*, *3*, and *Itgav* were observed, we could not identify apparent differences between CTRL-AO and BLM-AO. This result suggests that p53 does not directly induce the expression of these profibrotic TGF- β genes, at least in our assay. However, the direct p53 target genes identified by our ChIP-seq analysis included several TGF- β -related genes. We identified 329 genes that were the direct targets of p53 and upregulated in our RNA-seq data, including genes that have been reported to have a potential to induce or activate TGF- β signaling, such as *Areg*, *Cdkn1a*, *Ltbp2*, *Pdgfc*, and *Tgfa*. These data have been added to the revised Fig. 6 and Supplementary Table 4. These results have been described in the revised manuscript on Page 15, lines 9–18 and Page 16, lines 1–4 as follows:

“A time lag between p53 activation (day 7) and profibrotic TGF- β -related genes (day 9) prompted us to determine whether TGF- β isoforms and integrins are direct targets of p53. Therefore, we performed ChIP-seq analysis using an anti-p53 antibody for BLM-treated alveolar organoids to determine the direct targets of p53. We observed peaks upstream p53 target genes, such as *Cdkn1a* (Fig. 6A). However, there were no apparent differences between control- and BLM-treated-alveolar organoids in the binding of p53 to TGF- β isoforms and integrins (Fig. 6A), suggesting that p53 does not directly induce these profibrotic factors. However, the direct p53 target genes identified by our ChIP-seq analysis included several TGF- β -related genes (Fig. 6B). We identified 329 genes that were the direct targets of p53 and upregulated in our RNA-seq data, including genes that have been reported to have a potential to activate TGF- β signaling, such as *Areg*²⁹, *Cdkn1a*³⁰, *Ltbp2*³¹, *Pdgfc*³², and *Tgfa*³³ (Fig. 6C and Supplementary Table 4). These data suggest that p53 indirectly upregulates profibrotic factors by inducing these candidate factors.”

Fig. 5E

E GSE132910

Sftpc-creER; Sin3a-flox/flox; Rosa-mTmG

Fig. 6

A AT2 from WT → Day 0 → 6 → 7 ✓ ChIP-seq for TP53 using alveolar organoids

BLM

-Was the TGFbeta treatment of organoids for conditioned media swap done on organoids without fibroblasts?

The only evidence for TGFbeta being sufficient to bypass p53 is conditioned media swap experiments. In these experiments organoids, which have both epithelial and mesenchymal cells) were treated with TGFbeta and conditioned media was transferred to reporter fibs. What if exogenous TGFbeta activated fibroblast in organoids and this contributed to the CM effect because there is also an unseen autocrine loop in the fibroblasts?

Response:

We apologize for the confusion, but there were no fibroblasts in the culture for collecting the AO-conditioned medium. Kindly refer to the scheme in Fig. 6F. In this culture, alveolar organoids were treated with TGF-β for 24 h and washed before their co-culture with lung fibroblasts. Therefore, our findings mainly reflect AT2-derived TGF-β.

We agree that “an unseen autocrine loop” might exist in lung fibroblasts. Supporting this idea, fibroblasts seem to upregulate the expression of TGF-β ligands after exposure to TGF-β, although the expression of integrins was not increased. The TGF-β autocrine system may be commonly observed not only in AT2 cells but also in fibroblasts, which further promotes the pro-fibrotic status of these cells. These data have been added to the revised supplementary Fig. 6H (as shown below) and mentioned in the revised manuscript as follows (page 18, lines 8–13):

“In contrast, we found that treatment of fibroblasts with TGF-β also resulted in upregulated expression of TGF-β isoforms, particularly *Tgfb1* (Supplementary Fig. 6H), indicating that the TGF-β positive-feedback loop is a common cell response not only in AT2 cells but also in fibroblasts, which may contribute to maintaining the profibrotic state in both cells and accelerate the TGF-β signal interaction even without immune cells.”

Fig. 6H

-The authors hypothesize that p53-driven fibroblast-to-myofibroblast differentiation is mediated by activating latent TGF-beta through increased integrin alphaV-beta6 expression. They tested this hypothesis by adding a neutralizing antibody against integrin alphaV-beta6 in an AT2-fibroblast co-culture. However,

the antibody will block the integrin both on AT2 and fibroblasts. Can the authors show direct induction of integrin alphaV-beta6 by p53 activation in AT-lineage cells?

Response:

Thank you for your question. We would like to clarify that this culture experiment was performed without fibroblasts (Please see the revised Supplementary Fig.6G). Therefore, the antibody blocked integrin on AT2-lineage cells but not fibroblasts.

Supplementary Fig.6G

-Based on the cellchat analyses in figure 8H, the authors claim that epithelial cells can be affected by TGF β feedback loops. We believe more in-depth cell-cell communication analyses are necessary to reinforce this claim, both in terms of epithelial cell granularity as well as in the depth of analysis itself (e.g. quantification of the specific ABI-ABI TGF β signaling compared to other cell-cell TGF β signaling strength and the role of TGFBR2 herein).

Response:

Thank you for your excellent suggestion. We agree with you and have accordingly re-analyzed another scRNAseq dataset (GSE132771) of human IPF lungs. Using CellChat, we identified the presence of a TGF- β signaling loop in ABI cells, which would support our mouse findings even in human lungs. This result was added in the revised Fig. 9H and Supplementary Fig. 8D.

Fig. 9H

Supplementary Fig. 8D

-Other small comments: In the manuscript, the authors use a Acta2-DsRed mouse line to investigate fibroblast-to-myofibroblast differentiation. However, scRNASeq studies have shown that myofibroblast identity is mostly characterised by Cthrc1 and Tnc, rather than Acta2. Moreover, cthrc1high myofibroblasts were shown to have rather limited Acta2 expression. The authors need to show more evidence that the DsRed+ cells are indeed myofibroblasts. Similarly, the authors show Acta2 mRNA levels in figure 5E, suppl Fig 1 and suppl fig 3, which is insufficient to prove myofibroblast identity.

Response:

We agree with this comment. We have accordingly performed the suggested analyses and found that *Cthrc1* expression was increased in cultured fibroblasts treated with the BLM-AO-supernatant, as well as in those treated with recombinant TGF- β s, PDGF, and CTGF. In contrast, *Tnc* expression was not significantly increased even after exposure to TGF- β . We have added the data to Fig. 5E, Supplementary Fig. 1C, and Supplementary Fig. 5A and 5F.

Fig. 5E

Supplementary Fig. 1C

Supplementary Fig. 5A and 5F

o The authors use western blot to show protein expression, however no quantification is provided. This should be done, including statistical analysis.

Response:

Thank you for your suggestion. We have accordingly quantified the results of western blotting and added the results to the relevant Figures (Fig. 1J, Supple. Fig. 3E, Fig. 4C, and Fig. 8C). For data regarding p53-cKO AT2 cells, it was impossible to quantify the bands because p53 and p21 were not detected at all.

o The terms AT2-lineage cells and AT2 cells are both used in the manuscript in the context of the organoid and mouse models, leading to confusion. (This comment obviously does not apply to the section on human AT2 cells)

Response:

We have accordingly revised the term as appropriate throughout the manuscript.

Reviewer 3

In this manuscript the authors describe the effects of p53 and TGFbeta in eliciting non-inflammatory pulmonary fibrosis. Using genetically engineered mice, and organoid cultures as experimental systems they assess the role of damaged AT2 cells as critical mediators of fibrogenesis. Although much of the work confirms previous studies on idiopathic lung fibrosis the study does add new insights. It illustrates that no major role for immune cells has to be invoked in IPF although a further enhancing effect by factors produced by macrophages, neutrophils or other infiltrating immune cells is not excluded. They also show that the autocrine TGFbeta signaling in damaged AT2 cells plays a critical role in this process as knocking out the *Tgfr2* receptor in AT2 cells abrogates the fibrotic response in lung fibroblasts. The conclusions are supported by detailed and well-performed genetic experiments. However, there are a few issues the authors should address to make the manuscript suitable for publication in Nature Comm.^[1]

-In the first part of the manuscript the authors put a lot of emphasis on p53 signaling as a critical factor instructing AT2 cells to initiate the fibrotic process. However, other groups have performed more detailed analyses how p53 might more directly affect gene expression relevant for this process (e.g. as shown for the effects seen for PATS, Kobayashi et al. Nature Cell Biol. 2020). Therefore, this can be described more succinct with more extensive reference to recent publications describing the well published p53-TGFbeta-integrin alphaVbeta6 axis.

Response:

Thank you for your comment. In this revision, we have added new data regarding p53-TGF- β -ItgaV/Itgb6 axis using ChIP-seq (revised Fig. 6) and a reference of EMBO reports 2010;11:97–105 (doi:10.1038/embor.2009.276) that mentioned the ability of TGF- β to upregulate integrins. Furthermore, we also reanalyzed the data of single-cell RNA sequencing of Sin3a conditional KO within AT2 cells (GSE132910; Yao C, et al. AJRCCM 2021), in which p53-enriched AT2-lineage cells show upregulation of TGF- β 1 and ItgaV/Itgb6 expression (revised supplementary Fig. 5E). This information has been added as follows Page 14, lines 12–14): “These data are consistent with a previous study on Sin3a-cKO mice, in which p53-enriched AT2-lineage cells show upregulation of *Tgfb1*, *Tgfb3*, *Itgav*, and *Itgb6* (Supplementary Fig. 5E).¹⁵”

Supplementary Fig. 5E

E GSE132910

-Given the prominent role ascribed to p53 signaling in this study, the western blots to support the induction of p53 and Pser15-p53 (figures 3c, 4c, and their respective supplementary figures) are not very convincingly showing this. One would expect a more robust induction than is shown. Are other inducers of senescence such as INK4a/b, Mdm2, p19Arf, and p63 more convincingly induced, as senescence is a prominent feature of Bleomycin treatment AT2 cells as well as of PATS.

Response:

We agree that our western blots for p-p53 (Ser15) were not clear and not convincing. To overcome this issue, we performed western blot with an anti-acetylated-p53 (Ac-p53/Lys379) antibody and found that acetylated-p53 is significantly upregulated in bleomycin-treated AT2 cells, reflecting p53 signaling activation. In the revised manuscript, we replaced the data of Ac-p53 in Fig. 3C, Fig. 8, and Supplementary Fig. 3E.

In addition, to address the reviewer's comment, we sought senescence inducers other than p21, but only the p53-p21 axis was consistently confirmed as a senescence phenotype in bleomycin-treated AT2 or bleomycin-treated *Tgfb2*-null AT2 (Fig. 3C and 8C). These results suggest that downstream targets of p53 depend on the context. Thus, other inducers of senescence, such as INK4a/b, Mdm2, p19Arf, and p63, may not be the major target of p53 in AT2 cells, which is supported by our ChIP-seq analysis with anti-p53 (Fig. 6). These ChIP-seq analysis results have been described in the revised manuscript on Page 15, lines 9–18 and Page 16, lines 1–4 as follows:

“A time lag between p53 activation (day 7) and profibrotic TGF- β -related genes (day 9) prompted us to determine whether TGF- β isoforms and integrins are direct targets of p53. Therefore, we performed ChIP-seq analysis using an anti-p53 antibody for BLM-treated alveolar organoids to determine the direct targets of p53. We observed peaks upstream p53 target genes, such as *Cdkn1a* (Fig. 6A). However, there were no apparent differences between control- and BLM-treated-alveolar organoids in the binding of p53 to TGF- β isoforms and integrins (Fig. 6A), suggesting that p53 does not directly induce these profibrotic factors.

However, the direct p53 target genes identified by our ChIP-seq analysis included several TGF- β -related genes (Fig. 6B). We identified 329 genes that were the direct targets of p53 and upregulated in our RNA-seq data, including genes that have been reported to have a potential to activate TGF- β signaling, such as *Areg*²⁹, *Cdkn1a*³⁰, *Ltbp2*³¹, *Pdgfc*³², and *Tgfa*³³ (Fig. 6C and Supplementary Table 4). These data suggest that p53 indirectly upregulates profibrotic factors by inducing these candidate factors.”

Fig. 3C

Supplementary Fig. 3E

Fig. 8C

-The experiments illustrating a role for TGF-beta in the AT2 lung fibroblast interaction are more revealing. However, it remains unclear for the reader to what extent TGF-beta itself can induce the full fibrotic response achieved with the conditioned medium of bleomycin-treated AT2 cells. Although the authors tested a series of SASP factors produced by BLM treated AT2 cells, and conclude that TGFbeta is the only factor inducing the fibrotic response (as also depicted in the explanatory diagram) it remains unclear whether the maximal fibrotic response in lung fibroblast induced by the lowest concentration of TGFbeta

can be further enhanced by the addition of conditioned medium from BLM treated AT2 cells, as this would point to additional (not tested) factors that can synergize with TGFbeta in augmenting fibrosis.

Response:

Thanks for pointing out an important point that we had not tested. To determine whether the addition of conditioned medium can synergize TGF-β-induced myofibroblast differentiation, we treated fibroblasts with the organoid supernatant with or without a low dose of TGF-βs (10 pg/mL). Accordingly, we observed only additive, and not synergistic, roles between TGF-β and organoid-supernatant in the expression of fibrotic genes. These data were added to supplementary Fig. 5F and described in the manuscript as follows (Page 15, lines 4–8):

“Furthermore, to evaluate additional BLM-treated AT2-derived profibrotic factors, we cultured fibroblasts and treated them with organoid supernatant with or without low-dose TGF-βs (10 pg·mL⁻¹). As shown in Supplementary Fig. 5F, only additive, but not synergistic, changes in the expressions of myofibroblast-related genes were observed.”

Although we cannot completely exclude the possibility of other profibrotic factors secreted from BLM-treated AT2, TGF-β appears to be the most critical factor.

Supplementary Fig. 5F

REVIEWERS' COMMENTS

Reviewer #1 (Remarks to the Author):

The authors have addressed my concerns. I have not further questions.

Reviewer #2 (Remarks to the Author):

We thank the authors for their thorough revision of the manuscript and the complete point-by-point response to our comments.

We feel all comments have been addressed properly and limitations acknowledged. We do not have any further comments and feel the manuscript is ready for acceptance.

Reviewer #3 (Remarks to the Author):

The authors have adequately addressed my concerns and the manuscript is much improved. The revised MS can be accepted.